# Accelerated Learning with Linear Temporal Logic using Differentiable Simulation

## Abstract

To ensure learned controllers comply with safety and reliability requirements for reinforcement learning in real-world settings remains challenging. Traditional safety assurance approaches, such as state avoidance and constrained Markov decision processes, often inadequately capture trajectory requirements or may result in overly conservative behaviors. To address these limitations, recent studies advocate the use of formal specification languages such as linear temporal logic (LTL), enabling the derivation of correct-by-construction learning objectives from the specified requirements. However, the sparse rewards associated with LTL specifications make learning extremely difficult, whereas dense heuristic-based rewards risk compromising correctness. In this work, we propose the first method, to our knowledge, that integrates LTL with differentiable simulators, facilitating efficient gradient-based learning directly from LTL specifications by coupling with differentiable paradigms. Our approach introduces soft labeling to achieve differentiable rewards and states, effectively mitigating the sparse-reward issue intrinsic to LTL without compromising objective correctness. We validate the efficacy of our method through experiments, demonstrating significant improvements in both reward attainment and training time compared to the discrete methods.

## 1 Introduction

The growing demand for artificial intelligence (AI) systems to operate in a wide range of environments underscores the need for systems that can learn through interaction with their environments, without relying on human intervention. Reinforcement learning (RL) has emerged as a powerful tool for training controllers to perform effectively in uncertain settings with intricate, high-dimensional, and nonlinear dynamics. Recent advances in RL have enabled attainment of high-performance controllers in a variety of applications [1], such as robotic arm control [2], hand manipulation [3], legged locomotion [4, 5], navigation in crowded spaces [6], and robot-assisted surgery [7]. Despite the promising results in controlled environments, deploying learned controllers in real-world systems –where malfunctioning can be costly or hazardous– requires not only high performance but also strict compliance with formally specified safety and reliability requirements. Therefore, ensuring that learned controllers meet these critical specifications is essential to fully realize the potential of AI systems in real-world applications. Safety in learning is often modeled with constrained Markov decision processes (MDPs) [8–12], where the accumulated cost must be within a budget. However, additive cost functions may not reflect real-world safety, as assigning meaningful costs to harms is challenging. Alternative approaches define safety by avoiding unsafe states or actions [13–27], which is simpler than designing cost functions. However, this may result in overly conservative policies and could not capture complex trajectory-level requirements.

Recently, researchers have explored specifying RL objectives using formal languages, which explicitly and unambiguously express trajectory-based task requirements, including safety and liveness properties. Among these, linear temporal logic (LTL) has gained particular popularity [28–50] due the automaton-based memory it offers, which ensures history-independence and makes it especially suitable for long-horizon tasks unlike other languages such as signal temporal logic (STL). Specifying desired properties in LTL inherently prevents mismatches between the intended behavior and the behavior learned through reward maximization–one of the most well-known safety challenges in AI [51]. Although these methods are proven to define the correct RL objectives, the sparse logical

rewards make learning extremely difficult, as obtaining a nonzero reward often requires significant exploration. Denser LTL-based rewards provided through heuristics might accelerate learning [38]; however, if not carefully designed, they can compromise the correctness of the objective and misguide exploration depending on the environment, ultimately reducing learning efficiency. In this work, we address the challenges of scalable learning with correct objectives for long-horizon learning tasks. We adopt LTL as the specification language, leveraging the intuitive high level language and the automaton-based memory it provides. Unlike prior methods, our approach harnesses gradients from differentiable simulators to facilitate efficient learning directly from LTL specifications while preserving the correctness of the objectives. Our contributions can be summarized as follows:

- We propose, to the best of our knowledge, the first approach that accelerates learning from LTL specifications using differentiable simulators. Our approach effectively mitigates the inherent issue of the sparse rewards without sacrificing the expressiveness and correctness that LTL provides.

- We introduce soft labeling techniques for continuous environments that yield probabilistic $\varepsilon$-actions and transitions within the automata derived from LTL, which ensures the differentiability of rewards and states with respect to actions.

- Through a serious of experiments, we demonstrate that our approach enables successful learning from LTL specifications in robotic systems, whereas traditional non-differentiable approaches fail to achieve feasible learning.

## 2   Related Work

**Safe RL.**   One common perspective in Safe RL defines safety as the guarantee on the cumulative costs over time within a specified safety budget, which is often modeled using constrained MDPs and has been widely studied [8–12], relying on additive cost functions and budgets, which may not adequately capture safety in many scenarios. In practice, it is often difficult to assign unambiguous scalar costs reflecting trade-offs between different harmful situations [52]. Another approach defines safety in terms of avoiding unsafe states and focuses on preventing or modifying unsafe actions via shielding or barrier functions [13–27], which only require identification of unsafe states and actions and often easier than designing cost functions [53]; however, they can lead to overly conservative control policies [54]. Moreover, the requirements are often placed over trajectories, which could be more complex than simply avoiding certain states [55]. Our approach avoids these issues by employing LTL as the specifications language to obtain correct-by-construction RL objectives.

**RL with Temporal Logics.**   There is growing increasing in using formal specification languages to encode trajectory-dependent task objectives, especially those involving safety. LTL is widely used due to its expressiveness and well-defined semantics over infinite traces. There has been increasing interest in using formal specification languages to encode task objectives that are trajectory-dependent, particularly those involving safety requirements. LTL has emerged as a widely adopted formalism due to its expressiveness and well-defined semantics over infinite traces. Recent efforts [28–50] derives rewards from LTL specifications for RL, typically by translating LTL into limit-deterministic Büchi automata (LDBAs) and assigning rewards based on acceptance conditions. The memory structure provided by LDBAs supports long-horizon tasks better than alternatives such as STL, which require history-dependent rewards [56]. However, LTL rewards are often sparse and hinder learning. While heuristic-based dense rewards [38] attempt to address this, they risk misguiding exploration and compromising correctness. Our approach avoids these pitfalls by leveraging gradients from differentiable simulators to accelerate learning without sacrificing correctness.

**RL with Differentiable Simulators.**   Differentiable simulators enable gradient-based policy optimization in RL by computing gradients of states and rewards with respect to actions, using analytic methods [57–61] or auto-differentiation [62, 63]. While Backpropagation Through Time (BPTT) is commonly used [64–69], it suffers from vanishing or exploding gradients in long-horizon tasks as it ignores the Markov property of states [70]. To address this, several differentiable RL algorithms have been proposed [71, 72]. Short Horizon Actor-Critic (SHAC) [73] divides long trajectories into shorter segments where BPTT is tractable and bootstraps the remaining trajectory using the value function. Adaptive Horizon Actor-Critic (AHAC) [74] extends SHAC by dynamically adjusting the segment lengths based on contact information from the simulator. Gradient-Informed PPO [75] incorporates gradient information into the PPO framework in an adaptive manner. Our approach builds a differentiable, Markovian transition function for LTL-derived automata, making it compatible with all differentiable RL methods. Unlike prior STL-based efforts [76, 77], which rely on non-Markovian rewards and BPTT, our method supports efficient long-horizon learning with full differentiability.

## 3 Preliminaries and Problem Formulation

**MDPs.** We formalize the interaction between controllers with the environments as MDPs, which can be used for a wide range of robotic systems, including arm manipulation and legged locomotion.

**Definition 1.** *A (differentiable) MDP is a tuple $M = (S, A, f, p_0)$ such that $S$ is a set of continuous states; $A$ is a set of continuous actions; $f : S \times A \mapsto S$ is a differentiable transition function; $p_0$ is an initial state distribution where $p_0(s)$ denotes the probability density for the state $s$.*

For a given robotic task, the state space $S$ can be defined by the positions $\mathrm{x}$ and velocities $\dot{\mathrm{x}}$ of relevant objects, body parts, and joints. The action space $A$ may consist of torques applied to the joints. The transition function $f$ captures the underlying system dynamics and outputs the next state via computing the accelerations $\ddot{\mathrm{x}}$ by solving $\mathrm{M}\ddot{\mathrm{x}} = \mathrm{J}^{\mathrm{T}}\mathrm{F}(\mathrm{x}, \dot{\mathrm{x}}) + \mathrm{C}(\mathrm{x}, \dot{\mathrm{x}}) + \mathrm{T}(\mathrm{x}, \dot{\mathrm{x}}, a)$, for a given state $s = \langle \mathrm{x}, \dot{\mathrm{x}} \rangle \in S$ and action $a \in A$. Here, $\mathrm{F}$, $\mathrm{C}$, and $\mathrm{T}$ are, respectively, force, Coriolis, and torque functions that can be approximated using differentiable physics simulators.

**RL Objective.** In RL, a given policy $\pi : S \mapsto A$ is evaluated based on the expected cumulative reward (known as return) associated with the paths $\sigma := s_0 s_1 \ldots$ (sequence of visited states) generated by the Markov chain (MC) $M_\pi$ induced by the policy $\pi$. Specifically, for given a reward function $R : S \mapsto \mathbb{R}$, a discount factor $\gamma \in (0, 1)$, and a horizon $H$, the return of a path $\sigma$ from time $t \in \mathbb{N}$ is defined as $G_{t:H}(\sigma) = \sum_{i=t}^{H} \gamma^i R(\sigma[i])$. For simplicity, we denote the infinite-horizon return starting from $t = 0$ as $G_H(\sigma) := G_{0:H}(\sigma)$, and further drop the subscript to write $G(\sigma) := \lim_{H \to \infty} G_H(\sigma)$. The discount factor $\gamma$ reduces the value of future rewards to prioritize immediate ones: a reward received after $t$ steps, $R(\sigma[t])$, contributes $\gamma^t R(\sigma[t])$ to the return. The objective in RL is to learn a policy that maximizes the expected return over trajectories.

**Labels.** In robotic environments, we define the set of atomic propositions (APs), denoted by $\mathtt{A}$, as properties of interest that place bounds on functions of the state space. Formally, each AP takes the form $\mathtt{a} := \text{'}g(s) > 0\text{'}$, where $g : S \mapsto \mathbb{R}$ is assumed to be a differentiable function mapping a given state to a signal. For example, the function $g(\langle \mathrm{x}, \dot{\mathrm{x}} \rangle) := \dot{\mathrm{x}}_{\max}^2 - \dot{\mathrm{x}}_i^2$ can be used to define an AP that specifies that the velocity of the $i$-th robotic component must be below an upper bound $\dot{\mathrm{x}}_{\max}$. The labeling function $L : S \mapsto 2^{\mathtt{A}}$ returns the set of APs that hold true for a given state. Specifically, an AP $\mathtt{a} := \text{'}g(s) > 0\text{'}$ is included in the label set $L(s)$ of state $s$ – i.e., $s$ is labeled by $\mathtt{a}$ if and only if (iff) $g(s) > 0$. We also write, with a slight abuse of notation, $L(\sigma) := L(\sigma[0])L(\sigma[1]) \ldots$ to denote the trace (sequences of labels) of a path $\sigma$. Finally, we write $M^+ = (M, L)$ to denote a labeled MDP.

**LTL.** LTL provides a high-level formal language for specifying the desired temporal behaviors of robotic systems. Alongside the standard operators in propositional logic – negation ($\neg$) and conjunction ($\wedge$) – LTL offers two temporal operators, namely next ($\bigcirc$) and until ($\mathsf{U}$). The formal syntax of LTL is defined by the following grammar ([78]): $\varphi := \text{true} \mid \mathtt{a} \mid \neg\varphi \mid \varphi_1 \wedge \varphi_2 \mid \bigcirc\varphi \mid \varphi_1 \mathsf{U} \varphi_2$, $\mathtt{a} \in \mathtt{A}$. The semantics of LTL formulas are defined over paths. Specifically, a path $\sigma$ either satisfies $\varphi$, denoted by $\sigma \models \varphi$, or not ($\sigma \not\models \varphi$). The satisfaction relation is defined recursively as follows: $\sigma \models \varphi$; if $\varphi = \mathtt{a}$ and $\mathtt{a} \in L(\sigma[0])$ (i.e., $\mathtt{a}$ immediately holds); if $\varphi = \neg\varphi'$ and $\sigma \not\models \varphi'$; if $\varphi = \varphi_1 \wedge \varphi_2$ and $(\sigma \models \varphi_1) \wedge (\sigma \models \varphi_2)$; if $\varphi = \varphi_1 \mathsf{U} \varphi_2$ and there exists $t \geq 0$ such that $\sigma[t:] \models \varphi_2$ and for all $0 \leq i < t$, $\sigma[i:] \models \varphi_1$. The remaining Boolean and temporal operators can be derived via the standard equivalences such as eventually ($\Diamond\varphi := \text{true} \mathsf{U} \varphi$) and always ($\Box\varphi := \neg(\Diamond\neg\varphi)$).

**LDBAs.** Whether a path $\sigma$ satisfies a given LTL formula $\varphi$ can be automated by building a corresponding LDBA, denoted by $\mathcal{A}_\varphi$ that is suitable for quantitative model-checking of MDPs ([79]). An LDBA is a tuple $\mathcal{A}_\varphi = (Q, q_0, \Sigma, \delta, B)$ where $Q$ is a finite set of states; $q_0 \in Q$ is the initial state; $\Sigma = 2^{\mathtt{A}}$ is the set of labels; $\delta : Q \times (\Sigma \cup \{\varepsilon\}) \mapsto 2^Q$ is a transition function triggered by labels; $B \subseteq Q$ is the accepting states. An LDBA $\mathcal{A}_\varphi$ accepts a path $\sigma$ (i.e., $\sigma \models \varphi$), iff its trace $L(\sigma)$ induces an LDBA execution visiting some of the accepting states infinitely often, known as the Büchi condition.

**Control Synthesis Problem.** Our objective is to learn control policies that ensure given path specifications are satisfied by a given labeled MDP. In stochastic environments, this objective translates to maximizing the probability of satisfying those specifications. We consider specifications given as LTL formulas since LTL provides a high-level formalism well-suited for expressing safety and other temporal constraints in robotic systems–and, importantly, finite-memory policies suffice to satisfy LTL specifications [80]. We now formalize the control synthesis problem as follows:

**Problem 1.** *Given a labeled MDP $M^+$ and a LTL formula $\varphi$, find an optimal finite-memory policy $\pi_\varphi^*$ that maximizes the probability of satisfying $\varphi$, i.e., $\pi_\varphi^* := \underset{\pi \in \Pi}{\arg\max} \, \mathrm{Pr}_{\sigma \sim M_\pi^+} \{ \sigma \mid \sigma \models \varphi \}$, where $\Pi$ is the set of policies and $\sigma$ is a path drawn from the Markov chain (MC) $M_\pi^+$ induced by $\pi$.*

## 4 Accelerated Learning from LTL using Differentiable Rewards

In this section, we present our approach for efficiently learning optimal policies that satisfy given LTL specifications by leveraging differentiable simulators. We first define product MDPs and discuss their conventional use in generating discrete LTL-based rewards for reinforcement learning. We then introduce our method for deriving differentiable rewards using soft labeling, enabling gradient-based optimization while preserving the logical structure of the specifications.

**Product MDPs.** A product MDP is constructed by augmenting the states and actions of the original MDP with indicator vectors representing the LDBA states. The state augmentations serve as memory modes necessary for tracking temporal progress, while the action augmentations, referred to as $\varepsilon$-actions, capture the nondeterministic $\varepsilon$-moves of the LDBA. The transition function of the product MDP reflects a synchronous execution of the LDBA and the MDP; i.e., upon taking an action, the MDP moves to a new state according to its transition probabilities, and the LDBA transitions by consuming the label of the current MDP state.

**Definition 2.** *A product MDP* $\mathbf{M} = (\mathbf{S}, \mathbf{A}, \mathbf{f}, \mathbf{p_0}, \mathbf{B})$ *is of a labeled MDP* $M^+ = (S, A, f, p_0, \mathbb{A}, L)$ *with an LDBA* $\mathcal{A}_\varphi = (Q, \Sigma = 2^\mathbb{A}, \delta, q_0, B)$ *derived from a given LTL formula* $\varphi$ *such that* $\mathbf{S} = S \times \mathbf{Q}$ *is the set of product states and* $\mathbf{A} = A \times \mathbf{Q}$ *is the set of product actions where* $\mathbf{Q} = [0, 1]^{|Q|}$ *is the space set for the one-hot indicator vectors of automaton states;* $\mathbf{f} : \mathbf{S} \times \mathbf{A} \mapsto \mathbf{S}$ *is the transition function defined as*

$$\mathbf{f}(\langle s, \mathbf{q}^q \rangle, \langle a, \mathbf{q}^{q_\varepsilon} \rangle) := \begin{cases} \langle s', \mathbf{q}^{q'} \rangle & q_\varepsilon \notin \delta(q', \varepsilon) \\ \langle s', \mathbf{q}^{q_\varepsilon} \rangle & q_\varepsilon \in \delta(q', \varepsilon) \end{cases} \tag{1}$$

*for given* $s, s', \in S, a \in A$ *and the indicator vectors* $\mathbf{q}^q, \mathbf{q}^{q'}, \mathbf{q}^{q_\varepsilon} \in \mathbf{Q}$ *for* $q, q', q_\varepsilon \in Q$, *respectively, where* $s' := f(s, a)$ *and* $q' := \delta(q, L(s))$; $\mathbf{p_0}$ *is the initial product state distribution where* $p_0^\times(\langle s, \mathbf{q}^q \rangle)[q = q_0]$; $\mathbf{B} = \{\langle s, \mathbf{q}^q \rangle \in \mathbf{S} \mid q \in B\}$ *is the set accepting product states. A product MDP is said to accept a product path* $\boldsymbol{\sigma}$ *iff* $\boldsymbol{\sigma}$ *satisfies the Büchi condition, denoted as* $\boldsymbol{\sigma} \models \Box\Diamond\mathbf{B}$, *which is to visit some states in* $\mathbf{B}$ *infinitely often.*

By definition, any product path accepted by the product MDP corresponds to a path in the original MDP that satisfies the Büchi acceptance condition of the LDBA. Consequently, the satisfaction of the LTL specification $\varphi$ is reduced to ensuring acceptance in the product MDP. This reduces Problem 1 to maximizing the probability of reaching accepting states infinitely often in the product MDP:

**Lemma 1** (from Theorem 3 in [79]). *A memoryless product policy* $\boldsymbol{\pi}_\varphi^*$ *that maximizes the probability of satisfying the Büchi condition in a product MDP* $\mathbf{M}$ *constructed from a given labeled MDP* $M^+$ *and the LDBA* $\mathcal{A}_\varphi$ *derived from a given LTL specification* $\varphi$, *induces a policy* $\pi_\varphi^*$ *with a finite-memory captured by* $\mathcal{A}_\varphi$ *maximizing the satisfaction probability of* $\varphi$ *in* $M^+$.

**Discrete LTL Rewards.** The idea is to derive LTL rewards from the acceptance condition of the product MDP to be able train control policies via RL approaches. Specifically, we consider the approach proposed in [33] that uses carefully crafted rewards and state-dependent discounting based on the Büchi condition such that an optimal policy maximizing the expected return is also an objective policy $\boldsymbol{\pi}_\varphi^*$ maximizing the satisfaction probabilities as defined in Lemma 1, as formalized below:

**Theorem 1.** *For a given product MDP* $\mathbf{M}$, *the expected return for a policy* $\pi$ *approaches the probability of satisfying the Büchi acceptance condition as the discount factor* $\gamma$ *goes to 1; i.e.,* $\lim_{\gamma \to 1^-} \mathbb{E}_{\boldsymbol{\sigma} \sim \mathbf{M}_\pi}[G(\boldsymbol{\sigma})] = Pr_{\boldsymbol{\sigma} \sim \mathbf{M}_\pi}(\boldsymbol{\sigma} \models \Box\Diamond\mathbf{B})$; *if the return* $G(\boldsymbol{\sigma})$ *is defined as follows:*

$$G(\boldsymbol{\sigma}) := \sum_{t=0}^\infty R(\boldsymbol{\sigma}[t]) \prod_{i=0}^{t-1} \Gamma(\boldsymbol{\sigma}[i]), \quad R(\mathbf{s}) := \begin{cases} 1-\beta & \mathbf{s} \in \mathbf{B} \\ 0 & \mathbf{s} \notin \mathbf{B} \end{cases}, \quad \Gamma(\mathbf{s}) := \begin{cases} \beta & \mathbf{s} \in \mathbf{B} \\ \gamma & \mathbf{s} \notin \mathbf{B} \end{cases} \tag{2}$$

*where* $\prod_{i=0}^{-1} := 1$, $\beta$ *is a function of* $\gamma$ *satisfying* $\lim_{\gamma \to 1^-} \frac{1-\gamma}{1-\beta} = 0$, $R : \mathbf{S} \mapsto [0, 1)$ *and* $\Gamma : \mathbf{S} \mapsto (0, 1)$ *are state-dependent reward and the discount functions respectively.*

The proof can be found in [33]. The idea is to encourage the agent to repeatedly visit an accepting state as many times as possible by assigning a larger reward to the accepting states. Further, the rewards are discounted less in non-accepting states to reflect that the number of visitations to non-accepting states are not important. The LTL rewards provided this approach is that the rewards are

very sparse; depending on the environment and the structure of the automaton, the agent might need
to blindly explore a large portion of the state space before getting a nonzero reward, which constitutes
the main hurdle in learning from LTL specifications.

**Differentiable LTL Rewards.** We propose employing differentiable reinforcement learning (RL)
algorithms and simulators to mitigate the sparsity issue and accelerate learning. However, the standard
LTL rewards described earlier are not only sparse but discrete, rendering them non-differentiable
with respect to states and actions. This lack of differentiability primarily stems from two factors: the
binary state-based reward function and discrete automaton transitions. To address this challenge, we
introduce probabilistic "soft" labels. We start by defining the probability that a given AP, denoted as
$\mathtt{a} := \text{`} g(s) > 0\text{'}$, belongs to the label $L(s)$ of a state $s$. Formally, we define this probability as:

$$\Pr(\mathtt{a} \in L(s)) = \Pr(g(s) > 0) := h(g(s)) = \frac{1}{1 + \exp(-g(s))}. \tag{3}$$

Although we use the widely adopted sigmoid function here[1], any differentiable cumulative distribution
function (CDF) $h : \mathbb{R} \mapsto [0, 1]$ could be applied. Building upon these probabilities, we define the
probability associated with a label $l$ as follows:

$$\Pr(L(s) = l) = \prod_{\mathtt{a} \in l} \Pr(\mathtt{a} \in L(s)) \prod_{\mathtt{a} \notin l}(1 - \Pr(\mathtt{a} \in L(s))). \tag{4}$$

These probabilistic labels induce probabilistic automaton transitions, causing the controller to observe
automaton states probabilistically. Consequently, instead of modeling automaton states as determin-
istic indicator vectors in product MDPs, we represent them as probabilistic superpositions over all
possible automaton states. By doing so, we design differentiable transitions and rewards within the
product MDP. Let $f_L : S \times \mathbf{Q} \mapsto \mathbf{Q}$ denote the function that updates the automaton state probabilities
based on the LDBA transitions triggered by probabilistic labels, and let $\mathbf{q}$ denote the vector where
each element $\mathbf{q}_q$ is the probability of being in automaton state $q$, then we can formally define:

$$f_L(\langle s, \mathbf{q}\rangle) = \mathbf{q}' \;\; \text{where} \;\; \mathbf{q}'_{q'} = \sum_q \mathbf{q}_q \sum_{l \in L_{q,q'}} \Pr(L(s) = l) \;\; \text{and} \;\; L_{q,q'} := \{l \mid q' = \delta(q, l)\}. \tag{5}$$

Intuitively, the probability of transitioning to a subsequent automaton state $q'$ is computed by summing
probabilities across all current automaton states $q$ and labels $l \in L_{q,q'}$ capable of leading to state $q'$.
This computation can be efficiently done through differentiable matrix multiplication.

The remaining hurdle is the binary $\varepsilon$-actions available to the controller, which trigger $\varepsilon$-transitions in
the LDBA. Similarly to the soft labels approach, $\varepsilon$-actions can become differentiable by representing
the probabilities of the $\varepsilon$-transitions to be triggered. Let $f_\varepsilon : \mathbf{Q} \times \mathbf{Q} \mapsto \mathbf{Q}$ denote the function
updating automaton state probabilities based on the $\varepsilon$-action taken, and let $\mathbf{q}^\varepsilon$ denote the vector whose
elements indicate the probabilities of taking the $\varepsilon$-actions leading to the corresponding automaton
states, we then define:

$$f_\varepsilon(\mathbf{q}, \mathbf{q}^\varepsilon) = \mathbf{q}' \; \text{where} \; \mathbf{q}'_{q'} = \sum_{q \in Q_{\varepsilon,q'}} \mathbf{q}_q \mathbf{q}^\varepsilon_{q'} + \sum_{q \in \overline{Q}_{q',\varepsilon}} \mathbf{q}_{q'} \mathbf{q}^\varepsilon_q, \; Q_{\varepsilon,q'} := \{q \mid q' \in \delta(q, \varepsilon)\}, \; \overline{Q}_{q',\varepsilon} := \{q \mid q \notin \delta(q', \varepsilon)\}. \tag{6}$$

Conceptually, the probability of transitioning to automaton state $q'$ involves two scenarios: (the first
summation in (6)) the probability of moving to $q'$ via valid $\varepsilon$-transitions, and (the second summation
in (6)) the probability of remaining in $q'$ after trying to leave from $q'$ via nonexistent $\varepsilon$-transitions.
These vector computations can be efficiently performed in a differentiable manner.

We can formulate the complete transition function $\mathfrak{f}$ by composing $f_L$, $f_\varepsilon$, and $f$ as follows:

$$\mathfrak{f}(\langle s, \mathbf{q}\rangle, \langle a, \mathbf{q}^\varepsilon\rangle) := \langle f(s, a), f_L(\langle s, f_\varepsilon(\mathbf{q}, \mathbf{q}^\varepsilon)\rangle)\rangle. \tag{7}$$

This transition function first executes the $\varepsilon$-actions, then performs the LDBA transitions triggered by
state labels to update the automaton state probabilities, while applying the given action to update the
MDP states. The function $\mathfrak{f}$ is fully differentiable with respect to $s$, $\mathbf{q}$, $a$, and $\mathbf{q}^\varepsilon$. We can now obtain

---

[1]For the correctness of LTL, $\Pr(g(s) > 0)$ must be exactly 0 or 1 for values below or above certain thresholds.
In practice, this is not an issue, as overflow behavior of sigmoid ensures this condition is satisfied

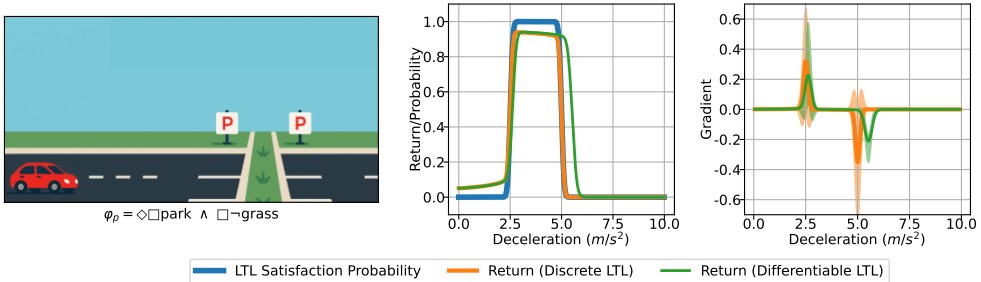

Figure 1: **LTL Returns and Derivatives.** *Left*: The parking scenario where the car must brake to stop in the parking area without entering the grass field ($\varphi_p$). *Middle*: LTL satisfaction probability and return estimates from discrete and differentiable LTL formulations as functions of deceleration. *Right*: LTL return gradients with respect to deceleration and their standard deviation. The key challenge in learning from LTL arises from slightly-sloped regions and sharp changes in the returns produced by discrete LTL rewards. Our *differentiable LTL* approach not only *smooths these abrupt changes but also enables the use of low-variance first-order gradient estimates essential for effective learning in slightly-sloped regions.*

a reward $\mathfrak{R} : \mathbf{Q} \mapsto (0, 1)$ and a discounting function $\mathfrak{D} : \mathbf{Q} \mapsto (0, 1)$ that are also differentiable with respect state and actions as follows:

$$\mathfrak{R}(\langle s, \mathbf{q} \rangle) := (1 - \beta) \sum_{q \in B} \mathbf{q}_q, \quad \mathfrak{D}(\mathbf{q}) := \beta \sum_{q \in B} \mathbf{q}_q + \gamma \sum_{q \notin B} \mathbf{q}_q \tag{8}$$

These differentiable reward, discounting and functions allow us to obtain first-order gradient estimates $\nabla^1_\psi J(\psi) := \mathbb{E}_{\sigma \sim M_{\pi_\psi}} \left[ \nabla_\psi G_H(\sigma) \right]$ which are known to exhibit lower variance compared to zeroth-order estimates [73]. Such first-order estimates can be effectively utilized by differentiable RL algorithms to accelerate learning. In the following example, we illustrate employing these lower-variance gradient estimates is particularly crucial when learning from LTL rewards.

**Parking Example.** Consider a parking scenario in which the vehicle starts with an initial velocity of $v_0 = 10$ m/s. The controller applies the brakes with a constant deceleration $a \in [0 \text{ m/s}^2, 10 \text{ m/s}^2]$ over the next 10 seconds, with the goal of bringing the car to rest inside the parking area. For safety, the vehicle must not enter the grass field before reaching the parking zone on the right-hand side. We formalize these requirements in LTL as $\varphi_p = \Diamond \Box \texttt{park} \land \Box \neg \texttt{grass}$ where the parking area and the grass field are defined as $\texttt{park} := (x > 10 \text{ m} \land x < 20 \text{ m}) \lor (x > 30 \text{ m} \land x < 40 \text{ m})$ and $\texttt{grass} := x > 20 \text{ m} \land x < 30$ m, respectively.

Figure 1 illustrates this task, including satisfaction probabilities, returns, and gradients with respect to deceleration. The satisfaction probability is 1 for deceleration values between 2.5 m/s$^2$ and 5.0 m/s$^2$, and 0 outside this range. The differentiable LTL returns closely match the discrete ones, except near the boundaries of the satisfaction region, where the differentiable version produces smoother transitions. This smoothness is particularly evident in the gradient plots. Although differentiable LTL rewards yield smoother return curves, learning remains challenging due to the small gradient magnitudes across most of the parameter space except near the satisfaction boundaries. For instance, in the region between 0.0 m/s$^2$ and 2.5 m/s$^2$, the returns increase with deceleration, but noisy gradient estimates can still lead the learner away from the satisfaction region. Therefore, obtaining low-variance gradient estimates is especially beneficial when learning from LTL, where most of the landscape requires sharper gradients for effective optimization.

## 5 Experiments

In this section, through simulated experiments, we show learning from the differentiable LTL rewards provided by our approach is significantly faster than learning from standard discrete LTL rewards.

**Implementation Details.** We implemented our approach in Python utilizing the PyTorch-based differentiable physics simulator dFlex introduced in [73]. We used an NVIDIA GeForce RTX 2080 GPU, 4 Intel(R) Xeon(R) Gold 5218 CPU cores, and 32 gigabytes memory for each experiment.

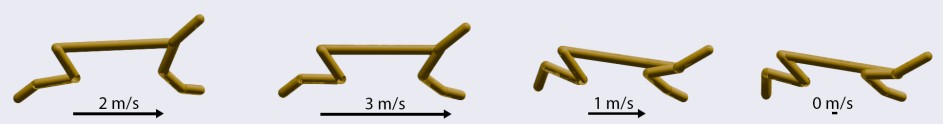

Figure 2: **Task Specification with LTL.** This figure illustrates a Cheetah policy learned by SHAC using differentiable rewards derived via our approach from the LTL formula $\varphi_{\text{legged}}$ (10), which specifies accelerating forward, stopping, and maintaining a safe tip-to-ground distance. Specifying the desired behaviors of robots using the high-level language LTL provides is an intuitive alternative to manually designing reward functions, which often require extensive domain expertise and risk unintended behaviors. Enabling learning directly from LTL unlocks new possibilities for robust, safe, and flexible robotic applications.

Specifically, we generate the automaton description using Owl [81] and parse it using Spot [82]. We then construct reward and transition tensors from the automata. We then compute the probabilities for each observations as explained in the previous section using a sequence of differentiable vector operations using PyTorch. Lastly, using the constructed transition and reward tensors, we update the automaton states and provide rewards. The overall approach is summarized in Algorithm 1.

**Baselines.** We use two widely adopted and representative state-of-the-art (SOTA) model-free RL algorithms as our baseline non-differentiable RL methods ($\partial$RLs): the on-policy Proximal Policy Optimization (PPO) [83] and the off-policy Soft Actor-Critic (SAC) [84]. For differentiable RL baselines ($\partial$RLs), we employ SHAC and AHAC, which, to the best of our knowledge, represent the SOTA in this category. For each environment and baseline, we adopted the tuned hyperparameters in [74].

---

**Algorithm 1** Differentiable RL with LTL

**Require:** MDP $M$, LTL formula $\varphi$, Policy $\pi_\psi$
   Derive LDBA $A_\varphi$ and APs $\mathbb{A}$ from $\varphi$
   Derive $\mathfrak{f}$ (7) and $\mathfrak{R}$, $\mathfrak{D}$ (8) from $A_\varphi$
   **while** True **do**
      Initialize $\mathbf{q}^{(0)} \sim A_\varphi$, $s^{(0)} \sim M$, $G \leftarrow 0$
      **for** $t = 1, 2, ..., H$ **do**
         Get action $\langle a, \mathbf{q}^\varepsilon \rangle \sim \pi_\psi(\langle s^{(t-1)}, \mathbf{q}^{(t-1)} \rangle)$
         Execute $\varepsilon$-action $\mathbf{q}' \leftarrow f_\varepsilon(\mathbf{q}, \mathbf{q}^\varepsilon)$
         Execute label transition $\mathbf{q}^{(t)} \leftarrow f_L(\langle s, \mathbf{q}' \rangle)$
         Execute MDP action $s^{(t)} \leftarrow f(s, a)$
         Compute reward $r \leftarrow \mathfrak{R}(\mathbf{q}^{(\mathbf{t})})$
         Update return $G \leftarrow G + \mathfrak{D}(\mathbf{q}) \cdot r$
      **end for**
      Train $\pi_\psi$ using differentiable return $G$
   **end while**

---

**Metric.** We evaluate performance in terms of the collected LTL rewards averaged over 5 seeds since they can serve as proxies for satisfaction probabilities. We considered two criteria: (1) the maximum return achieved and (2) the speed of convergence. To maintain consistency, we used differentiable LTL rewards across all baselines as, for non-differentiable baselines, we observed no performance difference between the differentiable and discrete LTL rewards.

**CartPole.** The CartPole environment consists of a cart that moves along a one-dimensional track, with a pole hinged to its top that can be freely rotated by applying torque. The system yields a 5-dimensional observation space and a 1-dimensional action space. The control objective is to move the tip of the pole through a sequence of target positions while maintaining the cart within a desired region as much as possible and ensuring the velocity of the cart always remains within safe boundaries. We capture these requirements in LTL as follows:

$$\varphi_{\text{cartpole}} = \underbrace{\square \text{'}|\texttt{cart\_vx}|<v_0\text{'}}_{\text{safety}} \wedge \underbrace{\square\lozenge\text{'}|\texttt{cart\_x}|<x_0\text{'}}_{\text{repetition}} \wedge \underbrace{\lozenge\left(\text{'}|\texttt{pole\_z-}z_0|<\Delta\text{'} \wedge \lozenge\text{'}|\texttt{pole\_z-}z_1|<\Delta\text{'}\right)}_{\text{reachability \& sequencing}}. \quad (9)$$

Here, $\texttt{cart\_x}$, $\texttt{cart\_vx}$, and $\texttt{pole\_z}$ represent the cart position, the cart velocity, and the pole height respectively. This formula demonstrates how LTL can be leveraged to encode both complex safety constraints and performance objectives. Specifically, we set $x_0 = 10$ m, $v_0 = 10$ m/s as boundaries, $z_0 = -1$ m, $z_1 = 1$ m as the target positions, and $\Delta = 25$ cm as the allowable deviation.

**Legged Robots.** We consider three legged-robot environments: Hopper, Cheetah, and Ant. The Hopper environment features a one-legged robot with 4 components and 3 joints, resulting in a 10-dimensional state space and a 3-dimensional action space. The Cheetah environment consists of a two-legged robot with 8 components and 6 joints, yielding a 17-dimensional state space and a 6-dimensional action space. The Ant environment includes a four-legged robot with 9 components and 8 joints, producing a 37-dimensional state space and an 8-dimensional action space. In all three environments, the control task requires always keeping the torso/tip of the robot above a critical safety height, maintaining a certain distance between the torso/tip and the critical height as often as

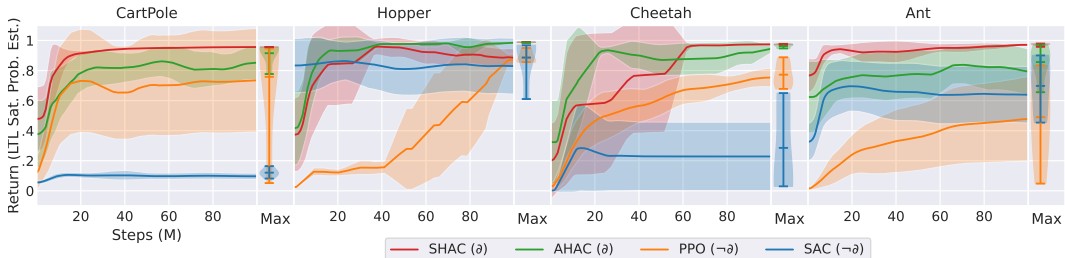

Figure 3: **Comparison Across Environments: Differentiable vs. Discrete LTL Rewards.** The wider plots show the learning curves of all baseline algorithms, while the narrower plots on the right display the maximum returns achieved after 100 M steps. All results are averaged over 5 random seeds, and the curves are smoothed using max and uniform filters for visual clarity. The reported returns, bounded between 0 and 1, serve as proxies for the probability of satisfying the LTL specifications. In all the environments algorithms utilizing **differentiable** LTL rewards (SHAC, AHAC) rapidly learn near-optimal policies, whereas those relying on discrete LTL rewards (PPO, SAC), display high variance, converge slowly, or getting stuck with sub-optimal/near-zero-return policies.

possible, and accelerating the robot forward, and then bringing the robot to a full stop. We formalize this task in LTL as follows:

$$\varphi_{\text{legged}} = \underbrace{\Box\text{`torso\_z}>z_0\text{'}}_{\text{safety}} \land \underbrace{\Box\Diamond\text{`torso\_z}>z_1\text{'}}_{\text{repetition}} \land \underbrace{\Diamond\big(\text{`torso\_vx}>v_1\text{'} \land \Diamond\text{`torso\_vx}<v_0\text{'}\big)}_{\text{reachability \& sequencing}}. \quad (10)$$

Here, torso_z and torso_vx denote the height and horizontal velocity of the robots. This formula captures several key aspects of LTL, including, safety, reachability, sequencing, and repetition. The values of $z_0$ and $z_1$ were chosen based on the torso height of each robot in their referential system. Specifically, we used $z_0 = -110$ cm, $z_1 = -105$ cm for Hopper; $z_0 = -75$ cm, $z_1 = -105$ cm for Cheetah; and $z_0 = 0$ cm, $z_1 = 5$ cm for Ant, where $z_0$ denotes the critical safety height and $z_1$ represents a safe margin above it. We set $v_1 = 1$ m/s, $v_1 = 3$ m/s, and $v_1 = 1.5$ m/s for Hopper, Cheetah, and Ant, respectively, reflecting movement speeds relatively challenging yet achievable for each of the robot. For deceleration, we set $v_0 = 0$ m/s for all the environments. An illustration of a policy learned from this specification for Cheetah is provided in Figure 2.

**Results.** Figure 3 presents our simulation results. Across all environments, $\partial$RL algorithms that leverage our differentiable LTL rewards consistently outperform $\not\partial$RL algorithms in terms of both maximum return achieved and learning speed from the LTL specifications.

*CartPole.* The safety specification induces an automaton with three states, each having 64 transitions–but only one of these transitions yields a reward. This extreme sparsity, even in a low-dimensional state space, severely hinders the learning process for $\not\partial$RLs, as shown in the leftmost plot of Figure 3. In contrast, $\partial$RL algorithms leverage the gradients provided by differentiable rewards, enabling them to efficiently learn policies that nearly satisfy the LTL specification. Specifically, $\partial$RLs converge to near-optimal policies (Pr>0.8) within just 20 M steps, whereas $\not\partial$RLs (SAC: all seeds; PPO: one seed) fail to learn any policy that achieves meaningful reward, even after 100 million (M) steps.

*Legged Robots.* As we move to environments with higher-dimensional state spaces–10, 17, and 37 dimensions for Hopper, Cheetah, and Ant, respectively–even relatively simple LTL specifications pose a significant challenge for $\not\partial$RLs. The automata derived from the LTL specifications in these environments consists of four states, each with 16 transitions, of which four transitions in the third state yield rewards. Reaching this state, however, requires extensive blind exploration of the state space, making it significantly hard for $\not\partial$RLs to learn optimal control policies. On the other hand, $\partial$RLs, guided by LTL reward gradients, quickly identify high-reward regions of the state space and learn effective policies.

For Hopper, $\partial$RLs converge to near-optimal policies (Pr>0.8) within 20 M steps, while PPO requires the full 100 M steps to converge, and one SAC seed gets trapped in a local optimum. For Cheetah, $\partial$RLs attain optimal performance (Pr>0.9), whereas PPO converges to a suboptimal policy even after 100 M steps, and SAC consistently fails by getting stuck in poor local optima. For Ant, $\partial$RLs again learn near-optimal policies rapidly, while $\not\partial$RLs converge only to suboptimal policies.

**Ablation Study.**    To isolate the impact of differentiability of LTL rewards from inherent environment properties, we conduct an ablation study comparing $\partial$RLs and $\not\partial$RLs under simplified LTL specifications. Specifically, we use reduced versions of the LTL formulas from our earlier experiments:

$$\varphi'_{\text{cartpole}} := \Diamond \text{`}|\text{pole\_z-}z_0|<\Delta\text{'} \tag{11}$$

$$\varphi'_{\text{legged}} := \Diamond \text{`torso\_vx}>v_1\text{'} \tag{12}$$

using $z_0 = -1$ m, $\Delta = 25$ cm for Cartpole, and $v_1 = 50$ cm/s for all the legged-robot environments. These simplified formulas yield one-state automata with 4 and 2 transitions, respectively, of which one is accepting. As such, they lack the complexity that makes learning from LTL challenging. Figure 4 presents the maximum returns obtained for these simplified specifications. Each of the baselines, regardless of differentiability, learns an optimal policy (Pr>0.9) for all the environments. However, when comparing these results to those in Figure 3, we observe only a minor performance drop for $\partial$RLs, whereas the performance of $\not\partial$RLs degrades dramatically—for some cases, from near satisfaction to complete failure—as LTL complexity increases. These results support our hypothesis that the performance advantage of $\partial$RLs over $\not\partial$RLs arises primarily from leveraging the differentiability of LTL rewards provided by our approach, rather than from environment-specific properties.

## 6    Discussion and Limitations

Our approach accelerates learning from LTL specifications by leveraging differentiable RL algorithms that utilize gradients provided by differentiable simulators. Therefore, the overall performance of our method is inherently influenced by the quality and efficiency of the underlying simulators and RL algorithms. For example, if a simulator provides poor gradient information or computes gradients slowly, the learning process will be significantly slowed down. Another issue is the reliance on hyperparameters. Although we adopt tuned hyperparameters from existing work, applying our approach to new environments may require additional hyperparameter tuning. A further challenge lies in the formalization of LTL specifications. While LTL offers a more intuitive and structured way to specify tasks compared to manual reward engineering, it still requires familiarity with formal logic and sufficient domain knowledge to define meaningful bounds. Finally, our method introduces an additional hyperparameter: the CDF used for probability estimation, which must also be tuned for optimal performance.

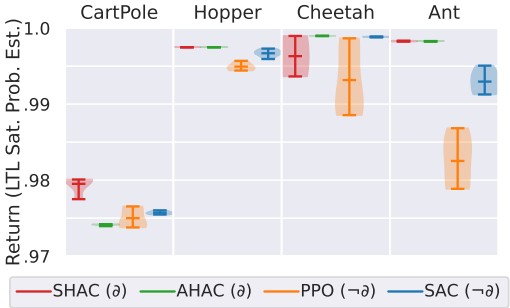

Figure 4: **Ablation Study for LTL.** The maximum returns obtained after 100 M steps for simplified LTL formulas (12), averaged over 5 seeds. Returns (0 to 1) indicate LTL satisfaction probabilities. Under these simpler specifications, both $\not\partial$RLs and $\partial$RLs successfully learn near-optimal policies. However, as shown in Figure 3, the *performance of discrete $\not\partial$RLs degrades dramatically with increasing LTL complexity—unlike differentiable $\partial$RLs, which maintain reasonable performance by leveraging the LTL rewards differentiability.*

## 7    Conclusion

In this work, we tackle the critical challenge of scalable RL for robotic systems under long-horizon, formally specified tasks. By adopting LTL as our specification framework, we ensure objective correctness and avoid the reward misspecification issues commonly encountered in conventional RL approaches. To overcome the learning inefficiencies caused by sparse logical rewards, we propose a novel method that leverages differentiable simulators, enabling gradient-based learning directly from LTL objectives without compromising their expressiveness or correctness. Our approach introduces soft labeling techniques that preserve the differentiability through the transitions of automata derived from LTL formulas, resulting in end-to-end differentiable learning framework. Through a series of simulated experiments, we demonstrate that our method substantially accelerates learning compared to SOTA non-differentiable baselines, paving the way for more reliable and scalable deployment of autonomous robotic systems in complex real-world environments.

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

# A    Appendix

Please check the supplemental material.

