# OpenReview forum: "Accelerated Learning with Linear Temporal Logic using Differentiable Simulation"
_NeurIPS.cc/2025/Conference — Submitted to NeurIPS 2025_

### Official Review · Reviewer_hBtv · 2025-06-17

**Clarity:** 2
**Significance:** 3
**Originality:** 3
**Rating:** 4
**Confidence:** 4

**Summary:**

Logical formalisms enable the specification of agents’ behaviors. For instance, they serve as building blocks for formally verifying agents and synthesizing agents’ controllers that guarantee satisfying the specification. In RL, the user needs to engineer a reward signal to align its intention with the induced agent’s behaviors, which is, in practice, very hard. Alternatively, one can use that logic to automatically derive a reward scheme for the agent. In particular, the authors consider LTL; in a nutshell, a return is assigned to each agent’s trajectory depending on its satisfaction of the LTL formula. However, traditional methods lead to a sparse reward scheme, which is not suitable for allowing the agent to learn to satisfy the user specification. The authors propose to use differentiable environments and labeling functions to allow the policy’s gradients to flow through both the environment and specification dynamics. The method relies on considering the labeling function as a probability distribution over the atomic proposition space instead of a deterministic function. Then, doing the product of the automaton translated from the formula with the environment leads to a differentiable environment that can be exploited to derive smooth, low-variance gradients of the policy.

**Questions:**

- How does the method compare to RL with usual reward schemes (e.g., those proposed in the original environments or reward machines; see e.g. [1])?
- Could you comment on the concerns raised about the function $g$ (see limitations)?
- Does the method work on (subclasses of) stochastic environments?

[1] LTL and Beyond: Formal Languages for Reward Function Specification in Reinforcement Learning. Alberto Camacho, Rodrigo Toro Icarte, Toryn Q. Klassen, Richard Valenzano, Sheila A. McIlraith, IJCAI 2019.

**Ethical Concerns:**

["NO or VERY MINOR ethics concerns only"]

**Final Justification:**

After having read the other reviews, I observe that we all noticed (1) the lack of guarantees linked to the introduction of the smooth labels, as well as (2) the lack of comparisons to previous methods (such as reward machines) in the evaluation.

In response to the weaknesses, limitations I raised, as well as my questions, the authors suggest the introduction of a new theorem as well as other non-negligible changes to the manuscript. Furthermore, the introduction of this new bound would probably be worth evaluating in the experiments section.

While the present paper is promising, it will need significant refactoring and polishing. For this reason, I maintain my original score.

**Limitations:**

- The method heavily relies on signal functions $g_a \colon S \to \mathbb{R}$ and thresholds such that the atomic proposition $a$ is part of the label iff $g_a(s) > 0$. On the other hand, the ‘intensity’ of this signal is used (heuristically) to define the probability of the atomic proposition being included in the label. I’m convinced (e.g., through the braking example) that this heuristic is of interest in some robotics applications. However, it is not clear to me how it is useful/sound in general, in particular for safety applications. For instance, hard thresholds might be relevant for safety, and it might be very difficult to design this function. One could, e.g., design a region that the robot is strictly forbidden to visit, whereas it is free to move within the rest of the state space.
- The authors restrict themselves to deterministic MDPs. If I am not mistaken, several stochastic transition functions (e.g., reparameterizable distributions) are differentiable. So, why not take such functions into account? The proposed method does not seem to require the MDP to be deterministic, as the resulting product is stochastic by introducing soft labeling.
- As acknowledged by the authors, the proposed methods shift from tuning a reward signal to tuning an LTL specification to make the agent learn, while requiring extra hyperparameter tuning. It would be great to include an analysis of how much this LTL formula tuning influences the results (in the appendix if space doesn’t permit it).

**Quality:**

3

**Strengths And Weaknesses:**

# Strengths
- In the tasks considered in the paper, the proposed approach empirically confirms the claims of the authors that differentiable LTL allows for avoiding high-variance gradient updates and alleviates the sparse rewards problem
- The idea of introducing soft labeling to allow differentiating the LTL formula (through its automata construction and generated rewards) is very promising.
- The experimental section is quite convincing and very detailed.
# Weaknesses
- The paper lacks accessibility; in my opinion, Section 3 lacks crucial details for understading the contribution. There is no real introduction to differentiable RL. Also, the formal logic specification part is extremely compact. For instance, for a noephyte, it is unclear what an LDBA actually does (what are \epsilon-actions, what do they represent, what does a path in the LDBA correspond to, how does the reward scheme of Theorem 1 work, what is beta, and how to choose it?). All those notions are essential to understanding the main contribution and would benefit from additional details and explanations.
- While being experimentally convincing, the method seems purely empirical and lacks theoretical support. For instance, how does the resulting probability signal relate to Theorem 1? Why is the sigmoid function a good fit for learning the rewards?
- Since the whole method relies on differentiable environments, to compute the gradients of the policy, I would have liked to see in Algorithm 1, called ‘differentiable RL with LTL’, how and where the gradients are updated. Currently, the algorithm is very generic and does not help understand that point. Note that I’m raising that issue because differentiable RL hasn’t been formally introduced anywhere in the paper.
- In Figure 3 (or in an additional figure), I think it could also be relevant to also plot the probability of the formula’s satisfaction with respect to the original rewards, i.e., the reward function provided in the considered environment. I think this is important to highlight the advantage of learning from LTL instead of classic reward schemes.
- The authors restrict themselves to deterministic MDPs and deterministic, stationary policies, and then claim that the goal of the agent is to optimize the expected return over trajectories. If everything is deterministic, we don’t need any expectation/probabilistic measures.
- Some limitations (listed below) are not acknowledged by the authors.

---

> ### Author Rebuttal · Authors · 2025-07-31
>
> We thank the reviewer for insightful feedback, valuable suggestions, and drawing attention to important aspects of our work. We provided detailed responses below and will revise the manuscript accordingly upon acceptance.
>
> ## Q1:
> The starting point of reward machine research is the construction of automata from LTL formulas specified as RL tasks [RM1]. Early work focused on the co-safe fragment of LTL [RM1] and LTL over finite traces (LTLf) [RM2], for which a deterministic finite automaton (DFA) can be constructed. These DFAs were later termed reward machines (RMs), as their accepting states can be used to assign positive rewards. Approaches targeting general LTL specifications instead use limit-deterministic Büchi automata (LDBAs). While structurally similar to RMs, LDBAs differ in their acceptance condition, which is based on infinite visitation of accepting states (suitable for continuous tasks), and thus require a distinct discounting scheme for correctness and convergence.
> In this work, we focus on making automaton-based reward signals differentiable, including those from DFAs and therefore RMs. When dealing with LDBAs, our approach additionally handles non-deterministic transitions and infinite visitation objectives. Importantly, our method can be immediately applied, without modification, to any RM constructed from co-safe LTL or LTLf) formulas using the atomic propositions defined in Section 2. As a result, our approach can directly accelerate learning with RMs. Furthermore, all existing techniques used with RMs, such as reward shaping [RM3], can also be integrated into differentiable RL. We note that while reward shaping does not affect the optimality of policies, it may lead to incorrect estimates of satisfaction probabilities. We hope our response provides sufficient explanation.
>
> [RM1] Icarte et al. "Teaching multiple tasks to an RL agent using LTL." AAMAS. 2018.
>
> [RM2] Camacho et al. "LTL and beyond: Formal languages for reward function specification in reinforcement learning." IJCAI. 2019.
>
> [RM3] Icarte et al. "Reward machines: Exploiting reward function structure in reinforcement learning." JAIR. 2022.
>
> ## Q2:
>
> By Theorem 1, the sum of discounted discrete LTL rewards approaches the actual satisfaction probabilities as $\gamma \to 1$, thereby adhering to hard safety constraints. The introduction of "softness" in the labels may introduce some slack, but this can be upper-bounded. We show that the maximum discrepancy between the discrete and differentiable values can be upper-bounded for a given tolerance parameter, as formalized in the theorem below.
>
> **Theorem 2:** Let $\varsigma$ be the tolerance on the signal bounds of atomic propositions, and let $p$ be the probability associated with $\varsigma$ (i.e., $p := \text{Pr}(\varsigma > 0) = h(\varsigma)$) as in (3). Let $G^\text{disc.}$ and $G^\text{diff.}$ denote the returns obtained via discrete and differentiable rewards, respectively. Then the maximum discrepancy between them is upper bounded as:
> $$
> |G^\text{disc.}(\sigma)-G^\text{diff.}(\sigma)| < \frac{1}{1+\frac{1-\beta}{(1-p)^{|\mathtt{A}|}}} = \frac{1}{1+\frac{1-\beta}{(1-h(\varsigma))^{|\mathtt{A}|}}}
> $$
> where $\beta$ is the discount factor for accepting states, as defined in (8). By linearity of expectation, this result immediately extends to the expected return (values) in stochastic environments; i.e, the upper bound above holds for $|\mathbb{E}[G^\text{disc.}(\sigma)]-\mathbb{E}[G^\text{diff.}(\sigma)]|$ where expectations are over trajectories drawn under any given policy.
>
> Please see Proof of Theorem 2 above under response to Reviewer ynyM.
>
> Now, given Theorem 2, we can capture the safety constraints via differentiable rewards with an approximation error that can be tuned by the tolerance parameter $\varsigma$ or the activation function $h$. For instance, using a hard sigmoid $h(x) = \max(0, \min(1,(\alpha x+1)/2))$ yields $p = 1$ when $\varsigma = 1/\alpha$, thus achieving equivalence. This introduces a trade-off between tolerance (hence the size of the the differentiable region) and the approximation error. We hope our response sufficiently addresses your concerns.
>
> ## Q3:
>
> Yes. We modeled the stochasticity of the environments through an initial state distribution ($p_0$ in MDPs, Section 3), since differentiable simulators typically have only deterministic step functions. Stochasticity can also be introduced by manually adding noise to the actuation signals, which can be integrated in a differentiable manner using the reparameterization trick as the reviewer pointed out. Our approach is capable of handling such environments, as our results hold in expectation over stochastic trajectories (Theorem 1, 2). In practice, we believe that as long as the noise isn't excessively high—which would pose a general challenge for RL—our differentiable approach will still lead to improved efficiency. We hope our response answers your question.
>
> ## W1:
> Due to the space limit, unfortunately, we had to move many details to Appendix. We include them below for your convenience.
>
> **Differentiable RL.** Differentiable simulators enable gradient-based optimization by analytically or automatically computing gradients of states and rewards with respect to actions. These simulators can be represented as differentiable transition functions $s_{t+1} = f(s_t, a_t)$, where $s_t$ and $a_t$ represent the state and action at time step $t$, respectively, and $s_{t+1}$ is the next state at time step $t+1$. In the context of RL, a common choice for the differentiable loss function is the negative of the return, defined as the sum of discounted rewards: $\mathcal{L} = -G_H(\sigma) = - \sum_{t=0}^H \gamma_t r_t$, where $H$ is the time horizon, $\sigma = s_0s_1\dots$ is the trajectory, $r_t$ is the reward at time step $t$, and $\gamma_t$ is the discount factor applied at that step. The backward pass then computes the gradients as follows:
> $$
> \frac{\partial G_H}{\partial s_t} = \frac{\partial G_H}{\partial s_{t+1}} \frac{\partial f}{\partial s_t}, \qquad
> \frac{\partial G_H}{\partial a_t} = \frac{\partial G_H}{\partial s_{t+1}} \frac{\partial f}{\partial a_t}.
> $$
> By chaining these gradients, the optimization updates propagate effectively throughout trajectories.In policy gradient RL over a finite horizon $H$, the goal is to find optimal parameters $\psi^\star = \mathrm{arg}\max_\psi J(\psi)$, such that $J(\psi) = \mathbb{E} \[G_H(\sigma)\]$ where expectation is over random trajectory drawn from the Markov chain induced by a policy $\pi$ parameterized by $\psi$.
> If a differentiable model is available, optimization can leverage first-order gradients:
> $\nabla_\psi^{\[1\]} J(\psi) = \mathbb{E} \[\nabla_\psi G_H(\sigma)\]$,
> or employ model-free zeroth-order gradients via the policy gradient theorem:
> $\nabla_\psi^{\[0\]} J(\psi) = \mathbb{E} \[G_H(\sigma) \sum_{t=0}^{H-1} \nabla_\psi \log \pi_\psi(a_t|s_t)\].$
> Both gradients can be approximated through Monte Carlo sampling:
> $\hat{\nabla_\psi}{\[1\]} J(\psi) = \frac{1}{N} \sum_{i=1}^N \nabla_\psi G_H(\sigma^{(i)})$ and $\hat{\nabla_\psi}^{\[0\]} J(\psi) = \frac{1}{N} \sum_{i=1}^NG_H(\sigma^{(i)}) \sum_{t=0}^{H-1} \nabla_\psi \log \pi_\psi(a^{(i)}_t|s^{(i)}_t)$ respectively.
>
> **LDBA Updates**. Please see the LDBA Updates paragraph under Q2 in our response to Reviewer ynyM; (also see the supp. file for details of LDBAs).
>
> **Rewarding and Discounting.** Visiting an accepting state requires a separate discount factor $\beta$ to ensure the correctness of the objective, namely, making the sum of discounted rewards accurately reflect the satisfaction probability. The core idea is to structure the rewards so that the cumulative reward from repeatedly visiting an accepting state converges to 1. For example, suppose an accepting state is visited repeatedly and a reward of $1 - \beta$ is provided at each time step. Then, the cumulative return is given by: $G = (1-\beta) + \beta(1-\beta) + \beta^2(1-\beta) + \dots = 1$. On the other hand, as long as accepting states are visited repeatedly, visits to non-accepting states do not violate the acceptance condition and thus should be disregarded. Since non-accepting states have no associated rewards, one might initially consider omitting discounting altogether in these states. However, doing so prevents convergence. Therefore, non-accepting states must still be discounted slightly, using a discount factor $\gamma$ that is significantly closer to 1 than $\beta$.
> We hope our response provides sufficient clarification.
>
> ## W2:
> Please see our detailed response to Q2. We hope it provides satisfactory explanation.
>
> ## W3:
> We believe our extended discussion in W1 addresses this concern as well.
>
> ## W4:
> Classic reward schemes are typically handcrafted for specific tasks and therefore cannot be directly analyzed in relation to general LTL formulas. That said, automatically deriving rewards from LTL specifications is significantly easier than manually designing classic rewards, which often requires detailed knowledge of system dynamics and balancing trade-offs between competing objectives.
>
> ## W5:
> Our discussion regarding stochasticity under Q3. We hope it provides sufficient explanation.
>
> ## L1 and L2:
> Please see our response to Q1 and Q2.
>
> ## L3:
> The choice of the activation function $h$ in $g$, along with its hyperparameters, can influence the performance of learning from LTL specifications. We intend to perform these experiments as much as possible and include the results in the manuscript before the camera-ready version upon acceptance.

---

> > ### Comment · Reviewer_hBtv · 2025-08-05
> >
> > Thank you for your responses to my questions and for reacting to the limitations and weaknesses I raised.
> >
> > Concerning your response to Q1, I think incorporating experimental comparisons would strengthen the evaluation of the approach.
> > For Q2, now that you have introduced a new theorem and bound, I think it could be interesting to include the latter as an additional metric for the evaluation as well in the final version of the paper.

---

> > > ### Author Response · Authors · 2025-08-06
> > > **THANK YOU!**
> > >
> > > Thank you again for all your insightful comments and questions.   We will include all the rebuttal response (inclusive of the added theorem and proof) in the final version of this paper.  We would very much appreciate if you may kindly consider raising your score to reflect on these clarifications and our notable contributions.
> > >
> > > Thank you again for your time and efforts in engaging with us on this review process.
> > >
> > > best,
> > >    The Authors

---

### Official Review · Reviewer_Ra9Y · 2025-06-22

**Clarity:** 2
**Significance:** 2
**Originality:** 3
**Rating:** 4
**Confidence:** 3

**Summary:**

This paper introduces a novel approach that integrates LTL specifications with differentiable physics simulators to enable efficient gradient based policy optimization directly from the logical objectives. The authors propose "soft labeling" technique that makes the normal discrete satisfaction conditions of LTL differentiable, generating comparatively smooth reward signals and automaton state transitions as functions of the agent's actions. Empirical results in simulated robotics show significantly faster convergence and higher attainment compared to traditional LTL-based methods.

**Questions:**

**1**, The rewards seem to be correlated with the probability of reaching accepting states in the LDBA. Depending on the reward design, will it possibly induce some reward hacking scenarios(eg. unexpected behaviors with high reward returns)? It would be interesting to provide some discussion in this direction.
**2**, If  we use the probabilistic/soft labels introduced in line 209 to 219, which also induced probabilistic automaton transitions, are we still dealing with LDBA? And also does Theorem 1 still holds under this setting?
**3**, What is the definition of $\psi$ defined in line 249? Is it controller parameters?
**4**, The motivation of ablation study conducting in simplified LTL is still unclear to the reviewer. Since here authors are using induced LTL formula not different set of formula how it can isolate the impact of differentiability of LTL rewards form inherent environment.

**Ethical Concerns:**

["NO or VERY MINOR ethics concerns only"]

**Final Justification:**

I think author's justification clarify my previous concerns and answer my questions, thus I decide to maintain my score.

**Limitations:**

This methods are limited for high dimensional continuous tasks due to the automaton generation process, but in general all LTL-based RL methods all have these kind of limitations. In general, this paper is technically solid with clear motivation and novelty. *

**Paper Formatting Concerns:**

No formatting concerns.

**Quality:**

3

**Strengths And Weaknesses:**

# Strengths
**1**, Literature survey is comprehensive and well written.\
**2**, Background introduction on labels, LTL, LDBA, and Control Synthesis are concise and easy to interpret for readers without model checking prior knowledge. \
**3**, The parking example helps reviewer conceptually better understand the rate of change of LTL satisfaction probability.\
**4**, The idea of integrating LTL with differential simulation and enable differentiability of discrete LTL objective/reward is novel.\
**5**, The differentiable rewards can provide more guidance in more rewarded direction, enabling faster convergence.


# Weakness
**1**, citation missing: LTL RL{[1,3,4],}; BF RL{[2]}.\
**2**, the explanation of how discounted function $\beta$ is defined is unclear to the reviewer and same for the discounting function $\mathfrak{D}$.\
**3**, The parking example is intuitive, but from figure 1 reviewer cannot tell the advantage in terms of low-variance.\
**4**, The reviewer think some other discrete LTL reward-based RL baselines are needed for comparisons. \
**5**, The reviewer think it is also necessary to demonstrate the safety rate/probability of satisfaction for both discrete and differentiable cases in more direct way, since intuitively the differentiability might introduce some slackness to the safety constraints.\



[1]Jackermeier, Mathias, and Alessandro Abate. "Deepltl: Learning to efficiently satisfy complex ltl specifications for multi-task rl." The Thirteenth International Conference on Learning Representations. 2025.
[2]Wang, Yixuan, et al. "Joint differentiable optimization and verification for certified reinforcement learning." Proceedings of the ACM/IEEE 14th International Conference on Cyber-Physical Systems (with CPS-IoT Week 2023). 2023.
[3]Camacho, Alberto, et al. "LTL and beyond: Formal languages for reward function specification in reinforcement learning." IJCAI. Vol. 19. 2019.
[4]Yalcinkaya, Beyazit, et al. "Compositional automata embeddings for goal-conditioned reinforcement learning." Advances in Neural Information Processing Systems 37 (2024): 72933-72963.

---

> ### Author Rebuttal · Authors · 2025-07-31
>
> We are grateful to the reviewer for their thoughtful feedback, constructive suggestions, and for drawing attention to important issues in our work. Below, we provide detailed responses and clarifications addressing the raised questions and concerns. We will update our manuscript upon acceptance accordingly.
>
>
> ## Q1:
> That is correct, the more frequently the accepting states of the LDBA are visited, the more rewards are accumulated. In our approach, the expected return (i.e., the cumulative discounted rewards) is proven to converge to the actual satisfaction probability in the limit as $\gamma \to 1$, given a sufficiently large horizon.
> That said, very long-horizon tasks typically require a discount factor $\gamma$ extremely close to 1. For example, a horizon of $H = 10000$ might necessitate $\gamma = 0.999999$ to maintain meaningful credit assignment over the trajectory. However, using such a high discount factor can significantly slow down convergence. In practice, smaller discount factors often lead to more efficient learning. The downside is that a smaller $\gamma$ places more emphasis on immediate rewards, increasing the risk of overlooking optimal policies that yield rewards later in the trajectory, such as those involving reaching a target state after many steps with higher certainty.
> We hope this response provides sufficient clarification.
>
> ## Q2:
> Yes, we still consider the satisfaction of the acceptance condition of LDBAs. One can interpret probabilistic LDBA updates as executions triggered by probabilistically superimposed trajectories. Theorem 1 still holds, but with an approximation error that can be controlled by a tolerance parameter and the choice of activation function. We formally state this as a theorem below:
>
> **Theorem 2:** Let $\varsigma$ be the tolerance on the signal bounds of atomic propositions, and let $p$ be the probability associated with $\varsigma$ (i.e., $p := \text{Pr}(\varsigma > 0) = h(\varsigma)$) as in (3). Let $G^\text{disc.}$ and $G^\text{diff.}$ denote the returns obtained via discrete and differentiable rewards, respectively. Then the maximum discrepancy between them is upper bounded as:
> $$
> |G^\text{disc.}(\sigma)-G^\text{diff.}(\sigma)| < \frac{1}{1+\frac{1-\beta}{(1-p)^{|\mathtt{A}|}}} = \frac{1}{1+\frac{1-\beta}{(1-h(\varsigma))^{|\mathtt{A}|}}}
> $$
> where $\beta$ is the discount factor for accepting states, as defined in (8). By linearity of expectation, this result immediately extends to the expected return (values) in stochastic environments; i.e, the upper bound above holds for $|\mathbb{E}[G^\text{disc.}(\sigma)]-\mathbb{E}[G^\text{diff.}(\sigma)]|$ where expectations are over trajectories drawn under any given policy.
>
> **Proof:** The maximum discrepancy between $G^\text{disc.}$ and $G^\text{diff.}$ occurs when all probabilistic transitions associated with soft labels yield positive differentiable rewards while their corresponding discrete rewards are zero (or vice versa). For a given tolerance $\varsigma$, the probability of incorrectly evaluating all atomic propositions under soft labels is $\rho := (1-p)^{|\mathtt{A}|}$, where $\mathtt{A}$ denotes the set of all atomic propositions defined in the LTL grammar.
> In this worst-case scenario, the differentiable return for a trajectory $\sigma$, where all such transitions lead to accepting states, is:
> $$
>     G^\text{diff.}(\sigma) \leq \sum_t^\infty \rho(1-\rho)^t\beta^t
>     = \frac{\rho}{1-(1-\rho)\beta} = \frac{\rho}{(1-\beta)+ \rho\beta}
>     = \frac{1}{1+\frac{1-\beta}{\rho}} = \frac{1}{1+\frac{1-\beta}{(1-p)^{|\mathtt{A}|}}}.
> $$
> Since $G^\text{disc.} = 0$, this expression provides the upper bound on the maximum discrepancy.
>
> Given Theorem 2, we can capture the Büchi acceptance condition via differentiable rewards with an approximation error that can be tuned by the tolerance parameter $\varsigma$ or the activation function $h$. For instance, using a hard sigmoid $h(x) = \max(0, \min(1,(\alpha x+1)/2))$ yields $p = 1$ when $\varsigma = 1/\alpha$, thus achieving equivalence. This introduces a trade-off between tolerance (hence the size of the the differentiable region) and the approximation error. We hope this provides a satisfactory explanation.
>
> ## Q3:
> Thank you for pointing this out. Here, $\psi$ denotes the set of trainable parameters, typically corresponding to the weights of the actor and critic networks in RL algorithms. We will make sure to explain this in detail in our paper.
>
>
> ## Q4:
> With the ablation studies, our goal is to highlight our core contribution: making learning for LTL specifications feasible through differentiable LTL rewards, which would otherwise be infeasible. As the reviewer pointed out, it is not possible to fully decouple the effects of environment dynamics from those of the task specification. However, these ablation studies demonstrate that there is no inherent limitation in the environments themselves that prevents learning from discrete rewards. They show that while learning with discrete rewards is possible for simple task specifications, it becomes dramatically more difficult as the complexity of the specification increases to include more challenging LTL properties. This underscores the necessity of using differentiable rewards for learning complex LTL tasks. We hope  this response provides sufficient clarification.
>
>
> ## W1:
> Thank you for bringing these articles to our attention. We will add these references.
>
>
> ## W2:
> Visiting an accepting state requires a separate discount factor to ensure the correctness of the objective, namely, making the sum of discounted rewards accurately reflect the satisfaction probability. The core idea is to structure the rewards so that the cumulative reward from repeatedly visiting an accepting state converges to 1. For example, suppose an accepting state is visited repeatedly and a reward of $1 - \beta$ is provided at each time step. Then, the cumulative return is given by: $G = (1-\beta) + \beta(1-\beta) + \beta^2(1-\beta) + \dots = 1$. On the other hand, as long as accepting states are visited repeatedly, visits to non-accepting states do not violate the acceptance condition and thus should be disregarded. Since non-accepting states have no associated rewards, one might initially consider omitting discounting altogether in these states. However, doing so prevents convergence. Therefore, non-accepting states must still be discounted slightly, using a discount factor $\gamma$ that is significantly closer to 1 than $\beta$.
>
> Due to the simultaneous execution of probabilistic transitions triggered by soft labels, the automaton occupies a superposition of states at any given timestep, represented by the vector $\mathbf{q}$. Because different automaton states require different discounting behaviors, we compute a superimposed discount factor, as specified by the function $\mathfrak{D}$, to account for this state-dependent discounting. We hope this response provides sufficient clarification.
>
>
> ## W3:
> The shaded areas in the rightmost plot of Figure 3 represent the standard deviation of the gradients. We observe that the gradient associated with differentiable rewards is significantly lower than that associated with discrete rewards, particularly around $5.0m/s^2$.
> We hope this helps.
>
> ## W4:
> Thank you for this suggestion.
> We would like to note that our primary objective in this work is to develop differentiable automaton-based rewards. Thus, our contribution can be measured by the improvement in learning efficiency achieved when transitioning from discrete to differentiable rewards, which is why we compare our approach directly to the discrete version. Our framework can easily be applied to any automaton-based reward method (such as reward machines), and similar comparisons can be performed between the standard discrete rewards and the differentiable rewards obtained via the soft labels introduced in our work.
> Upon acceptance, we will incorporate more discrete LTL algorithms, analyze the acceleration gained through differentiability via the soft labels we introduce, and include a comparative evaluation.
>
>
> ## W5:
> Thank you for pointing this out. It is, unfortunately, typically not feasible to compute satisfaction probabilities for continuous (infinite-horizon) tasks, such as safety, in highly nonlinear systems. However, we can employ a formal metric that indicates whether the system's progress complies with the specification, similar to the robustness score used in the online monitoring of STL specifications.
> We plan to include such results upon acceptance.

---

### Official Review · Reviewer_96Em · 2025-06-28

**Clarity:** 2
**Significance:** 2
**Originality:** 2
**Rating:** 4
**Confidence:** 4

**Summary:**

This paper addresses the problem of how to use differentiable simulators/differentiable reinforcement learning to learn in linear temporal logic problems. The reason this is hard to do is that formulating an LTL spec as an RL objective produces sparse discrete rewards which only appear in states that satisfy the spec. The paper produces smooth, differentiable rewards by calculating labels using a distance measure then turning this distance measure into a probability using a sigmoid function. The probability for a spec being satisfied is the product of the label probabilities. This approach is evaluated using two state of the art differentiable RL algorithms and compared against two standard RL algorithms on some basic control problems in simulation. The results show that the use of the differentiable rewards speed up learning and also find better policies i.e. policies that are more likely to satisfy the specification.

**Questions:**

1. How would this work in high dimensional settings?
2. Would you expect similar performance in stochastic domains?
3. How does this vary in practice from learning for for an STL spec?
4. Can you compare to reward machines?

**Ethical Concerns:**

["NO or VERY MINOR ethics concerns only"]

**Final Justification:**

Moved to borderline accept due to rebuttal. The clarification of relation to STL, removal of long horizon claims, and the one additional domain supported this. However I cannot go higher since this produces a large amount of work that will go unreviewed in the final version, and I think on balance the paper would be stronger with a further review round to look at all of this in context.

**Limitations:**

Yes

**Paper Formatting Concerns:**

84 references seems like a lot!

**Quality:**

3

**Strengths And Weaknesses:**

The key strength of the paper is proposing a dense reward structure that is correctly aligned with the LTL spec objective. This goes beyond existing work which generates dense rewards but without guarantees of creating optimal policies for the specification.  Whilst the idea of generating dense rewards is not original, this approach for propagating satisfaction probabilities for a dense reward is new, at least as far as I am aware. This is also quite significant since looks to allow scaling of RL for LTL specs, which is desirable to the community.

There are a few weaknesses of the paper.

I find there's not enough time spent comparing to or discussing existing work. Whilst there are a huge number of references, no individual reference is really explored in detail. Also there are no competing methods included in the evaluation as baselines. I was surprised not to see reward machines [37] compared to, since they seem a direct competitor to this method.

Also, given the way predicate satisfaction is formulated, I would have liked to see a deeper comparison to STL, since this appears equivalent to the taken approach to labelling.

I also think the evaluation domains don't really align with the robotics/long-horizon pitch of the paper. Cartpole and Deepmind Control are good starter control from state observation domains, but are reasonably easy for modern continuous control RL approaches. These are not long-horizon problems in robotics, even if specifying a sequence of objectives (move then stop) does go beyond a typical control horizon. They are also deterministic problems, and the specifications are over times which smoothly vary in state space anyway (i.e. velocity) which I assume makes it easier to learn probability maximising policies. In summary, I'd like to see the work evaluated in Richard domains as well as over more meaningful temporal specifications, particularly where those temporal variations may be more discrete than changes in velocity.

---

> ### Author Rebuttal · Authors · 2025-07-31
>
> We thank the reviewer for insightful comments, helpful suggestions, and pointing out the important issues regarding our paper. We answered and explained the questions and weaknesses below. We will update our manuscript upon acceptance accordingly.
>
>
> ## Q1:
> Learning from sparse rewards in high-dimensional settings is generally a challenging problem, primarily due to the extensive exploration required. In our paper, we demonstrate that our differentiable reward approach can significantly mitigate this issue. For example, as shown in Figure 3 for the Ant environment with a 37-dimensional state space, our method substantially improves learning efficiency. While learning from discrete LTL rewards in even higher-dimensional state spaces may potentially require billions of time steps, we believe that our differentiable approach can substantially alleviate this burden. We hope this answers your question.
>
> ## Q2:
> We modeled the stochasticity of the environments through an initial state distribution ($p_0$ in MDPs, as described in Section 3), since differentiable simulators typically have only deterministic step functions ($f$). Stochasticity can also be introduced by manually adding noise to the actuation signals, which can be integrated in a differentiable manner using the reparameterization trick. Our approach is capable of handling such environments, as our results hold in expectation over stochastic trajectories (Theorem 1). In practice, we believe that as long as the noise is not excessively high—which would pose a general challenge for RL—our differentiable approach will still lead to improved efficiency. We hope this addresses your question.
>
> ## Q3:
> STL additionally allows for the specification of real-time constraints; however, this comes at the significant cost of losing the compact memory mechanism provided by LTL. Specifically, evaluating STL satisfaction or robustness requires access to the full trajectory, as these metrics can only be computed after the trajectory ends. This necessitates storing entire trajectory histories, which directly violates the Markovian assumption fundamental to value-based RL techniques. Unlike LTL, this issue cannot be resolved by augmenting the state space with a compact memory representation derived from automata.
> There are two common approaches to address this challenge. The first is to augment the state space with the full trajectory history, but this leads to prohibitively large and impractical state spaces. For instance, with a horizon of $H = 1024$ used in our experiments, the state space for the Ant environment would be of size $1024 \times 37 = 37{,}888$. The second approach involves applying policy optimization over the action history using backpropagation through time (BPTT). However, due to the well-known exploding and vanishing gradient problems, gradients quickly deteriorate and become unreliable beyond roughly 100 time steps. This issue is further exacerbated in stochastic environments, where optimization becomes ineffective even over a small number of steps.
> Using intermediate STL robustness scores, an approach sometimes adopted to address the sparse reward problem in practice, does not resolve this hard non-Markovian problem of STL specifications. In fact, optimizing for the sum of such intermediate rewards can lead to incorrect RL objectives, as it does not necessarily correspond to maximizing the satisfaction of the STL specification.
> We hope our response provides sufficient information and clarification.
>
> ## Q4:
> The starting point of reward machine research is the construction of automata from LTL formulas specified as RL tasks [RM1]. Early work focused on the co-safe fragment of LTL [RM1] and LTL over finite traces (LTLf) [RM2], for which a deterministic finite automaton (DFA) can be constructed. These DFAs were later termed reward machines (RMs), as their accepting states can be used to assign positive rewards. Approaches targeting general LTL specifications instead use limit-deterministic Büchi automata (LDBAs). While structurally similar to RMs, LDBAs differ in their acceptance condition, which is based on infinite visitation of accepting states (suitable for continuous tasks), and thus require a distinct discounting scheme for correctness and convergence.
> In this work, we focus on making automaton-based reward signals differentiable, including those from DFAs and therefore RMs. When dealing with LDBAs, our approach additionally handles non-deterministic transitions and infinite visitation objectives. Importantly, our method can be immediately applied, without modification, to any RM constructed from co-safe LTL or LTLf) formulas using the atomic propositions defined in Section 2. As a result, our approach can directly accelerate learning with RMs. Furthermore, all existing techniques used with RMs, such as reward shaping [RM3], can also be integrated into differentiable RL. We note that while reward shaping does not affect the optimality of policies, it may lead to incorrect estimates of satisfaction probabilities.
> We hope this reponse provides sufficient clarification.
>
> [RM1] Icarte et al. "Teaching multiple tasks to an RL agent using LTL." AAMAS. 2018.
>
> [RM2] Camacho et al. "LTL and beyond: Formal languages for reward function specification in reinforcement learning." IJCAI. 2019.
>
> [RM3] Icarte et al. "Reward machines: Exploiting reward function structure in reinforcement learning." JAIR. 2022.
>
>
>
> ## W1: Existing Work
> Due to space limitations, we had to condense the related work section, which may have obscured some important details. We include our extended discussion below for your convenience and will update our paper to include more details.
>
> **RL for TL.** Initial attempts to combine LTL with RL  relied on model-based approaches. These methods required precise knowledge of the transition structure of the underlying MDPs to precompute accepting components of a product MDP constructed using a Deterministic Rabin Automaton (DRA) based on LTL specifications. This approach transformed satisfying temporal logic constraints into reachability problems solvable via RL. Despite providing probably approximately correct (PAC) guarantees, the complexity and frequent unavailability of accurate transition models limit their applicability, especially in deep RL contexts.
> To address these limitations, model-free RL methods for LTL emerged, such as those proposed in [RM(1-3)] for co-safe LTL and LTLf, which directly generate rewards from the acceptance conditions of automata derived from LTL without explicit knowledge of transition dynamics. The introduction of LDBAs [79], simplifying model checking by utilizing simpler Büchi conditions instead of Rabin conditions, further facilitated structured reward design with improved correctness [28] and stronger convergence guarantees [29].
> Researchers have also explored alternative temporal logics to generate informative reward signals. STL allows robustness scores to serve as rewards in finite-horizon tasks [56]. However, these scores typically depend on historical information, violating the Markov assumption and restricting their use in stochastic or value-based RL settings.
>
>
> ## W2: Baseline/RM Comparison
> Our objective in this work is to make automaton-based rewards differentiable. Therefore, our contribution can be measured by the acceleration in learning achieved by switching from discrete to differentiable rewards. This is why we compared our approach with the discrete version. Our framework can be readily applied to RMs, and a similar comparison can be made between standard discrete RM rewards and differentiable RM rewards obtained through the soft labels we introduce (e.g., by focusing on co-safe LTL or LTLf; also see our response to Q4). We will include such explicit comparisons upon acceptance.
>
>
> ## W3: STL Comparison
> Learning from STL rewards is generally not well-suited for the tasks with a horizon of $H = 1024$ considered in our paper (please see our response to Q3 as well), and therefore, we believe not suitable for direct comparison with LTL in our experiments. We hope our response provides clarification.
>
>
> ## W4: Long-Horizon Control
> We thank the reviewer for this comment. We will consider even longer horizons in our experiments as suggested. We also would like to emphasize that, in the environments we consider, only the forward movement objective, supported by carefully designed dense and intricate rewards, can be effectively solved by modern RL approaches. For example, the reward function used in the Ant environment is as complex as:
> reward = healthyreward\*ifhealthy() + wforward\*dx/dt - wcontrol\*$||$action$||^2_2$ - wcontrol\*$||$clip(Fcontact)$||^2_2$.
> General objectives without handcrafted rewards are difficult to learn for a horizon of $H = 1024$, the setting used in our experiments, even for state-of-the-art RL algorithms such as SAC and PPO, as demonstrated in our results.
>
>
> ## W5: Determinism
> Please see our response to Q2, where we also discuss how we can consider stochasticity in our setting.
>
>
> ## W6: More Discrete Temporal Specifications
> We thank the reviewer for this comment.
> We would like to note that, although differentiable simulators require the state space to be continuous, the positions and velocities can be highly nonlinear with respect to actions—often resembling discrete transitions—especially in contact-rich, legged-robot environments such as those used in our experiments. As a result, the The APs we use inherently exhibit highly nonlinear behavior with respect to actions. Additionally, any differentiable nonlinear function can be used for $g(s)$ in APs; however, the more nonlinear it is, the less reliable the gradients become. Nonetheless, this does not fundamentally affect our approach, as it is no different from having highly nonlinear transitions within the simulator itself.
>
>
> ## Formatting Concern:
> We will decrease the number of references.

---

> > ### Comment · Reviewer_96Em · 2025-08-01
> >
> > Thanks for your detailed response.
> >
> > To clarify a few of my points
> >
> > Long-Horizon Control: I think I was looking for something more qualitatively different here. Long-horizon control (at least in my world) is less about the size of H, but requiring different types of behaviours over the time horizon. This is often the case in robotic tasks where you need to pick an object, then move, then place, and repeat this for multiple objects.
> >
> > STL Comparison: I agree that scalability is an issue here. I guess I was less interested in whether RL for STL is a good method, but whether there's some more formal equivalence between the continuous signal problem you've created and directly working with an STL specification. This is particularly true given you can only apply your method to continuous domains where STL would be appropriate.
> >
> > Baselines/Domains: I think I'd still like to see comparisons to other reward shaping baselines (e.g. the shaping approach from RMs), to see how your acceleration compares to other approaches. And for completeness it would be good to see your method run on the continuous domains from the RM work. I realise that's a bit of a vague comment to make. I think your contribution is strong, but to argue for a the significance of general method like this, in this venue, I think the bar for empirical coverage should be set high.

---

> > > ### Author Response · Authors · 2025-08-04
> > >
> > > We would like to thank the reviewer again for carefully reviewing both our manuscript and response, and for providing insightful comments and suggestions that demonstrate a deep understanding of our work. Please find our further clarifications below:
> > >
> > > **Long-Horizon Control.** We now better understand the reviewer's point. It is indeed relatively straightforward to specify a sequence of tasks and repetitions using temporal logic. For example, an LTL formula such as $\varphi := \phi_1 \wedge \lozenge(\phi_2 \wedge \lozenge(\phi_3 \wedge \dots))$ enforces sequential execution of task specifications $\phi_1, \phi_2, \phi_3, \dots$. The trade-off, however, is an increase in the automaton size, which in turn raises both the dimensionality of the augmented state space and the sparsity of the rewards.
> > > Nonetheless, we believe our approach will significantly outperform discrete RL algorithms in such settings. As shown in our ablation studies, transitioning from a simple to a more complex LTL formula highlights the performance gap between discrete and differentiable approaches. Due to the extended training time required and the rebuttal deadline, we are unable to provide additional experimental results at this stage. However, if the paper is accepted, we will perform these experiments before the camera-ready submission and update the manuscript accordingly.
> > > Finally, we would like to note that another strategy for addressing long-horizon tasks is to train separately for each subformula $\phi_1, \phi_2, \phi_3, \dots$, assuming the formulas and environment support decoupling. However, this approach may lead to suboptimal local solutions.
> > >
> > > **STL Comparison.** We agree with the reviewer that there is indeed a formal equivalence. Specifically, we are interested in a subset of LTL formulas where atomic propositions (APs) are defined as bounds on continuous signals (or their functions). This subset overlaps with the class of STL formulas that do not have real-time constraints. One advantage of these formulas is that we avoid the inherently discrete APs permitted by general LTL formulas, thereby ensuring differentiability of the system. For example, instead of relying on a discrete low-battery indicator (e.g., `a := low_battery`), we define APs as continuous thresholds (e.g., `a := battery_level < 20%`). Another advantage is that abstracting away the real-time constraints inherent in general STL formulas enables us to employ an automata-based compact memory mechanism rather than storing the entire trajectory history, substantially improving learning efficiency. Finally, we would like to emphasize that working directly with continuous signals facilitates the online monitoring of robustness scores, which can be used for evaluating learned policies, a point we will certainly include in our manuscript upon acceptance.
> > >
> > > **Baselines/Domains** We understand the reviewer’s concern. We are currently running experiments for a continuous domain from the RM work (RM3). Once we obtain the results, we will compare them with those in RM3 and post here.

---

> > > > ### Comment · Reviewer_96Em · 2025-08-04
> > > >
> > > > **Long-Horizon Control** That makes sense. I'm not sure you need to run the experiments for the revised paper. Instead I would be happy to see the claim about long-horizons removed, since I don't think removing that claim affects your contribution. That said, I am curious about performance on the kinds of task you outline, since they are (for me at least), a very natural form of TL specification for robot behaviour, but I don't need this for the review.
> > > >
> > > > **STL Comparison** You've highlighted the equivalence, but doesn't this mean that RL approaches for STL problems should be considered as baselines or points of comparison, or at least should be discussed in more detail in the related work.  It also sounds like your subset is equivalent to Truncated Linear Temporal Logic from [1]. Is that fair?
> > > >
> > > > **Baselines/Domains** I look forward to the results :)
> > > >
> > > > [1] X. Li, C. -I. Vasile and C. Belta, "Reinforcement learning with temporal logic rewards," 2017 IEEE/RSJ International Conference on Intelligent Robots and Systems (IROS), Vancouver, BC, Canada, 2017, pp. 3834-3839, doi: 10.1109/IROS.2017.8206234. keywords: {Trajectory;Robots;Robustness;Semantics;Learning (artificial intelligence);Heuristic algorithms},

---

> > > > > ### Author Response · Authors · 2025-08-05
> > > > >
> > > > > Thank you for this constructive feedback.
> > > > >
> > > > > **Long-Horizon Control.** We will make sure to remove our claim about long-horizon control. Instead, we will emphasize the automata-based efficient memory mechanism perspective of our work, particularly for an extended number of time steps. Additionally, we will certainly consider the execution of a sequence of subtasks and aim to include these results in our paper.
> > > > >
> > > > > **STL Comparison.** We will ensure to expand the discussion of STL in our related work. We would like to emphasize that, although STL formulas without real-time constraints are syntactically equivalent to the LTL formulas we consider, STL significantly differs semantically. Specifically, STL places greater emphasis on the quantitative robustness score computed over the entire trajectory and does not aim to construct compact, automata-based memory. This distinction also holds true for TLTL from [1]; and as the reviewer pointed out, TLTL  is syntactically equivalent; but, similarly focuses on robustness scores. Due to this non-Markovian characteristic of STL (also see our response to Q3), we were unable to obtain meaningful experimental results with STL for the tasks considered in our paper. We hope this clarification adequately explains why we did not use STL as a baseline.
> > > > >
> > > > > **Baselines/Domains.** We have obtained results for the Cheetah environment described in [RM3]. We compare our results with those shown in Figure 10 of [RM3]. Specifically, we utilized the same reward machines described in `task 1` and `task 2`, making them differentiable using our proposed approach. Subsequently, we employed the SHAC algorithm to learn policies. Our differentiable approach outperformed all methods presented in Figure 10.
> > > > >
> > > > > | Steps ($1000$) |        task 1        |     task 2     |
> > > > > | :------------: | :------------------: | :------------: |
> > > > > |     $500$      | $7476.7 \pm 3516.5$  | $10.9 \pm 3.5$ |
> > > > > |     $1000$     | $12180.0 \pm 1899.6$ | $16.6 \pm 1.4$ |
> > > > > |     $1500$     | $11894.5 \pm 2776.9$ | $18.6 \pm 1.7$ |
> > > > > |     $2000$     | $13694.7 \pm 3244.6$ | $19.4 \pm 1.9$ |
> > > > > |     $2500$     | $14478.1 \pm 2659.3$ | $21.0 \pm 2.0$ |
> > > > > |     $3000$     | $15413.2 \pm 2548.8$ | $21.1 \pm 1.9$ |
> > > > >
> > > > > Thank you again for your review and suggestions; we believe these changes improve the quality of our paper.

---

> > > > > > ### Author Response · Authors · 2025-08-06
> > > > > > **Thank you!**
> > > > > >
> > > > > > Dear Reviewer 96 Em,
> > > > > >
> > > > > > Thank you again for your excellent questions and discussion with us.   We will include all the rebuttal response (inclusive of the added explanation and the additional experimental results above) in the final version of this paper.  We would very much appreciate if you may kindly consider raising your score to reflect on these added clarifications and notable contributions.
> > > > > >
> > > > > > Thank you again for your time and efforts in engaging with us on this review process.
> > > > > >
> > > > > > best,
> > > > > >    The Authors

---

> > > > > > > ### Comment · Reviewer_96Em · 2025-08-06
> > > > > > >
> > > > > > > You're welcome, it's interesting work. You have definitely addressed some of my concerns. I will raise my score to borderline accept. I think the two things I'd want to see in a future ML conference submission (in addition to the things discussed above) are one or two more appropriate baseline methods, and results on more domains where there are RM results or similar.

---

### Official Review · Reviewer_ynyM · 2025-07-03

**Clarity:** 1
**Significance:** 2
**Originality:** 2
**Rating:** 4
**Confidence:** 3

**Summary:**

This paper addresses the challenge of sparse rewards in reinforcement learning with linear temporal logic (LTL) specifications by introducing a novel differentiable framework. Traditional methods translate LTL into automata and provide sparse, discrete rewards, which are difficult to optimize. The proposed method transforms this process into a differentiable one by introducing probabilistic “soft” labels for atomic propositions and modeling transitions through a probabilistic automaton. This allows the construction of a product MDP with fully differentiable state transitions and reward structures, enabling the use of first-order gradient-based optimization algorithms.

The method is implemented using differentiable simulators and evaluated using both traditional and differentiable RL algorithms across standard benchmarks like CartPole, Hopper, Cheetah, and Ant. The experiments show that differentiable RL methods using this reward structure outperform traditional methods in convergence speed and policy quality, especially for complex LTL specifications.

**Questions:**

1.	Could you provide a theoretical guarantee or approximation result for the differentiable LTL rewards?
 Theoretical guarantees (e.g., Theorem 1) only apply to the discrete formulation. Does a similar result hold in expectation for the differentiable case?
2.	Please define or explain key notations more clearly (e.g., q_\epsilon, \sigma[i], \sigma[t:], and f_L).
 Several notations are introduced without adequate definitions or intuition, especially in Section 4. A short paragraph walking through the probabilistic automaton update step (Equation 5) would be helpful.

3.	Can you explain whether and how the gradient-based learning framework respects the Büchi acceptance condition probabilistically?
 While differentiable rewards are defined, it’s unclear how they preserve the infinite visitation condition (\always \eventually B) in practice or theory. Do the smooth reward functions approximate this in some probabilistic sense over finite horizons?

*The authors have answered my questions satisfactorily.*

**Ethical Concerns:**

["NO or VERY MINOR ethics concerns only"]

**Final Justification:**

The authors have answered my questions satisfactorily. As a result, I have increased my score.

**Limitations:**

- Have the authors acknowledged limitations and potential negative societal impact?
     Yes

**Paper Formatting Concerns:**

None.

**Quality:**

2

**Strengths And Weaknesses:**

Strengths
- The idea of integrating differentiable reinforcement learning with LTL-based specifications is, to the best of my knowledge, novel and timely. With increasing interest in structured reward functions and formal methods in RL, this paper contributes a natural progression by introducing a differentiable framework grounded in formal logic other than STL.
- The approach leverages soft labeling and probabilistic automaton state representations to construct end-to-end differentiable reward structures, addressing the longstanding challenge of reward sparsity in LTL-guided learning.
- The experiments are extensive, covering both low- and high-dimensional robotic control tasks. The results convincingly demonstrate the advantage of differentiable LTL rewards in terms of convergence and final policy quality.
- The ablation study strengthens the paper by isolating the contribution of differentiability from environmental factors.

Weaknesses
- There are multiple issues with mathematical clarity and presentation:
1. The definition of policies as \pi : S \to A implies memoryless policies, yet the paper later discusses finite-memory policies without offering sufficient background or formal definitions.
2. The notation M used in line 109 (transition dynamics) is not properly defined.
3.	\sigma is defined as a path, but inconsistently used — e.g., line 112 uses it as a single instance, and expressions like \sigma[i], \sigma[t:] are used without explanation.
4.	The set of atomic propositions (AP) is denoted in both bold and normal font without clarification (e.g., line 127).
5.	The state q_{\epsilon} is introduced in Equation (1) but is never defined or intuitively explained, leaving its role ambiguous.
6.	The function f_L (Equation 5) is described in dense notation without sufficient elaboration, making its mechanics difficult to follow.
7.	Line 74 (“growing increasing”) seems to have typos and is not readable.
8.	The theoretical results provided (e.g., Theorem 1) only apply to the discrete reward formulation, and no similar guarantees are provided for the differentiable LTL rewards introduced later in the paper. This weakens the formal justification of the core contribution.

---

> ### Author Rebuttal · Authors · 2025-07-31
>
> We appreciate the reviewer’s thoughtful comments, constructive suggestions, and for highlighting key issues in our paper. We have addressed and clarified the questions and concerns as detailed below. We will update our manuscript upon acceptance accordingly.
>
>
> ## Q1:
>
> We can show that the maximum discrepancy between the discrete and differentiable values can be upper-bounded for a given tolerance parameter, as formalized in the theorem below. Since, by Theorem 1, the discrete values converge to the satisfaction probabilities, the bound is also valid in the limit for the actual satisfaction probabilities.
>
> **Theorem 2:** Let $\varsigma$ be the tolerance on the signal bounds of atomic propositions, and let $p$ be the probability associated with $\varsigma$ (i.e., $p := \text{Pr}(\varsigma > 0) = h(\varsigma)$) as in (3). Let $G^\text{disc.}$ and $G^\text{diff.}$ denote the returns obtained via discrete and differentiable rewards, respectively. Then the maximum discrepancy between them is upper bounded as:
> $$
> |G^\text{disc.}(\sigma)-G^\text{diff.}(\sigma)| < \frac{1}{1+\frac{1-\beta}{(1-p)^{|\mathtt{A}|}}} = \frac{1}{1+\frac{1-\beta}{(1-h(\varsigma))^{|\mathtt{A}|}}}
> $$
> where $\beta$ is the discount factor for accepting states, as defined in (8). By linearity of expectation, this result immediately extends to the expected return (values) in stochastic environments; i.e, the upper bound above holds for $|\mathbb{E}[G^\text{disc.}(\sigma)]-\mathbb{E}[G^\text{diff.}(\sigma)]|$ where expectations are over trajectories drawn under any given policy.
>
> **Proof:** The maximum discrepancy between $G^\text{disc.}$ and $G^\text{diff.}$ occurs when all probabilistic transitions associated with soft labels yield positive differentiable rewards while their corresponding discrete rewards are zero (or vice versa). For a given tolerance $\varsigma$, the probability of incorrectly evaluating all atomic propositions under soft labels is $\rho := (1-p)^{|\mathtt{A}|}$, where $\mathtt{A}$ denotes the set of all atomic propositions defined in the LTL grammar.
> In this worst-case scenario, the differentiable return for a trajectory $\sigma$, where all such transitions lead to accepting states, is:
> $$
>     G^\text{diff.}(\sigma) \leq \sum_t^\infty \rho(1-\rho)^t\beta^t
>     = \frac{\rho}{1-(1-\rho)\beta} = \frac{\rho}{(1-\beta)+ \rho\beta}
>     = \frac{1}{1+\frac{1-\beta}{\rho}} = \frac{1}{1+\frac{1-\beta}{(1-p)^{|\mathtt{A}|}}}.
> $$
> Since $G^\text{disc.} = 0$, this expression provides the upper bound on the maximum discrepancy.
>
>
> ## Q2:
> We include definitions of $\sigma$ and its fragments ($\sigma[t]$, $\sigma[:t]$, $\sigma[t:]$) in the context of RL objective below (please see the supplementary material for more details):
>
> **RL Objective**. In RL, a given policy $\pi:S^+\mapsto A$ is evaluated based on the expected cumulative reward (known as return) associated with the paths $\sigma := s_0s_1\dots$ (sequence of visited states) generated by the Markov chain (MC) $M_\pi$ induced by the policy $\pi$.
> We write $\sigma[t]$, $\sigma[:t]$, $\sigma[t:]$ for $s_t$, the prefix $s_0\dots s_t$ and the suffix $s_ts_{t+1}\dots$.
> For a given reward function $R: S^+ \mapsto \mathbb{R}$, a discount factor $\gamma \in (0, 1)$, and a horizon $H$, the return of a path $\sigma$ from time $t \in \mathbb{N}$, is defined as $G_{t:H}(\sigma) = \sum_{i=t}^{H} \gamma^{i} R(\sigma[:i])$.
> For simplicity, we denote the infinite-horizon return starting from $t = 0$ as $G_H(\sigma) := G_{0:H}(\sigma)$, and further drop the subscript to write $G(\sigma) := \lim_{H \to \infty} G_H(\sigma)$. We note that for Markovian reward functions ($R: S \mapsto \mathbb{R}$), memoryless policies ($\pi:S\mapsto A$) suffice.
> However, the tasks we consider require finite-memory policies (see the supplementary material). To address this, we reduce the problem of obtaining a finite-memory policy to that of learning a memoryless policy by augmenting the state space $S$ with memory states, as detailed in Section 4.
> The discount factor reduces the~value of future rewards to prioritize immediate ones: a reward received after $t$ steps contributes $\gamma^t R(\sigma[t])$ to the return. The objective in RL is to learn a policy that maximizes the expected return over trajectories.
>
> We include an explanation of automaton updates with soft labels and $\varepsilon$-actions as follows:
>
> **LDBA Updates**. In an LDBA, transitions triggered by labels are deterministic, so the current automaton state is represented by a discrete variable $q \in \mathbb{N}$. In our setting, however, the automaton observes all possible labels simultaneously, each with an associated probability. Consequently, the current state becomes a probability distribution over possible states, represented by a vector $\mathbf{q}$.
> Similarly, instead of taking a single $\varepsilon$-action (as in the discrete case, where it corresponds to a transition to a specific automaton state), our framework allows the agent to ``take a probability distribution'' over $\varepsilon$-actions. Each component of this action vector corresponds to a potential transition to a different automaton state. This probabilistic formulation captures uncertainty and enables smoothness and gradient-based learning.
>
>
> ## Q3:
> In the literature, it is established that the discrete reward formulation captures the Büchi acceptance condition in the limit as the discount factor $\gamma \to 1$. For finite-state MDPs, there always exists a sufficiently large discount factor $\gamma < 1$ for any given LTL specification. For infinite number of MDP states, though we do not have such guarantees in theory (i.e., $\gamma$ required could be arbitrarily close to $1$); there usually exists a practical discount factor. Our setting has these limitations as well, as our differentiable rewards are crafted based on these discrete rewards.
>
> Now, given Theorem 2, we can similarly capture the Büchi acceptance condition via differentiable rewards with an approximation error that can be tuned by the tolerance parameter $\varsigma$ or the activation function $h$. For instance, using a hard sigmoid $h(x) = \max(0, \min(1,(\alpha x+1)/2))$ yields $p = 1$ when $\varsigma = 1/\alpha$, thus achieving equivalence. This introduces a trade-off between tolerance (hence the size of the the differentiable region) and the approximation error.
>
>
>
> ## W1:
> We provide a formal definition and explanation of finite-state machines below. (also see our response to Question 2 for a broader explanation). We also have a similar detailed explanation of finite-memory policies in the supplemantary file. Thank you for pointing this out, we understand the confusion and will update our paper accordingly.
>
> A *finite-memory policy* for an MDP $M$ is defined as a tuple $\pi = (\mathfrak{M}, \mathfrak{m}_0, \mathfrak{T}, \mathfrak{a})$, where:
> - $\mathfrak{M}$ is a finite set of modes (memory states);
> - $\mathfrak{m}_0 \in \mathfrak{M}$ is the initial mode;
> - $\mathfrak{T} : \mathfrak{M} \times S \times \mathfrak{M} \rightarrow [0,1]$ is a probabilistic mode transition function such that for any current mode $\mathfrak{m}$ and state $s$, the probabilities over next modes sum to 1, i.e., $\sum_{\mathfrak{m}' \in \mathfrak{M}} \mathfrak{T}(\mathfrak{m}' \mid \mathfrak{m}, s) = 1$;
> - $\mathfrak{a} : \mathfrak{M} \times S \times A \rightarrow [0,1]$ is a probabilistic action selection function that assigns a probability to each action $a$ given the current mode $\mathfrak{m} \in \mathfrak{M}$ and state $s \in S$.
>
> A finite-memory policy acts as a finite-state machine that updates its internal mode (memory state) as states are observed, and specifies a distribution over actions based on both the current state and mode. The action at each step is thus selected according not only the current state but also the current memory state of the policy.
> In contrast to standard definitions of finite-memory policies, which typically assume deterministic mode transitions, this definition permits probabilistic transitions between modes.
>
>
> ## W2:
> The symbol $\mathtt{M}$ denotes the mass matrix. Thank you for catching this oversight.
>
>
> ## W3:
> Thank you for pointing this out. Due to the space limitation, we had to move the path definitions and notations to supplementary material. We understand the confusion, and will make sure to include brief explanations in the paper. Please also see our response to Question 2.
>
>
> ## W4:
> This discrepancy appears to be a PDF rendering issue. We confirm that AP refers to atomic propositions and that $\mathtt{A}$ denotes the corresponding set.
>
>
> ## W5:
> We represent the $\varepsilon$-action being taken with an integer $q_{\epsilon}$, which denotes the automaton state reached via the $\varepsilon$-transition associated with that action. We also have a more detailed description of LDBAs and $\varepsilon$-transitons in the supplementary file. We could not explain in detail due to space limitations, but we understand this could be confusing and we will update the manuscript accordingly.
>
>
> ## W6:
> We included an intuitive explanations of LDBAs via soft labels in our response to Question 2. We hope this helps.
>
>
> ## W7:
> Thank you for catching this. It was a leftover sentence fragment.
>
>
> ## W8:
> We address this by introducing Theorem 2, which provides a formal approximation bound between discrete and differentiable rewards (please also see our response to Question 1). We hope this helps strengthen the formal justification of our core contribution.

---

> > ### Author Response · Authors · 2025-08-06
> > **Thank you**
> >
> > Dear Reviewer ynyM:
> >
> > As the discussion period is winding down, we wonder if our response addressed your concerns.   In particular, we added a proof for the Theorem 2, per your request.   We hope that our theoretical analysis and proof, supported by additional experimental results (see below), clarify any remaining concern you may have.   Please kindly let us know if our rebuttal response to your comments and other reviewer's questions assure you the notable contributions of this work and the technical rigor+significance of this paper.   If you have no further question, we would truly appreciate if you may kindly consider raising your score to acknowledge these exchanges and clarifications on issues you suggested.
> >
> > Thank you.
> >
> > best,
> >    The Authors

---

> > ### Comment · Reviewer_ynyM · 2025-08-06
> >
> > Thank you for addressing my questions. I have no further comments and will accordingly update my score.

---

> > > ### Author Response · Authors · 2025-08-06
> > >
> > > Thank you for your support!   Very much appreciated!

---

### Note · Authors · 2025-08-14

We sincerely thank the reviewers for their detailed and constructive feedback. Addressing the reviewers’ concerns, as well as engaging in discussions during the rebuttal, has helped us clarify and strengthen both the theoretical and empirical contributions.

Our work presents the ***first*** end-to-end differentiable reinforcement learning (RL) framework for learning controllers from linear temporal logic (LTL) specifications in differentiable simulation environments; further narrowing the gap between formal methods and deep RL. We introduce a method that makes LTL-based automaton rewards differentiable in continuous control domains, enabling gradient-based optimization directly from formal specifications. Our approach significantly accelerates learning, ***achieving up to twice the returns of the baselines*** across diverse experiments in complex, nonlinear, contact-rich across diverse settings such as the 37-dimensional, 8-DoF quadruped Ant environment, where even standard RL struggles without handcrafted reward shaping.

During the rebuttal, we ***extended our experiments to include reward machines*** (which generalize to *co-safe LTL and LTLf* without modification) and ***introduced Theorem 2***, which provides a formal, tunable bound on the discrepancy between *discrete and differentiable* LTL returns. Together with Theorem 1, this fully establishes the connection between Büchi acceptance and discrete/differentiable LTL rewards in both *deterministic* and *stochastic* settings. We also expanded our discussion of background material, formalization details, stochastic environments, related work, and other formal specification languages, such as signal temporal logic. We will ensure that these clarifications, along with the missing citations and minor corrections, are included in the final version.

We believe this paper lays the foundation for a new paradigm in learning from high-level linear temporal logic specifications in continuous domains, achieving both theoretical guarantees and empirical acceleration. By making automaton-based rewards differentiable, we enable seamless integration of symbolic task specifications into modern gradient-based RL pipelines. This capability, not addressed by existing methods, opens new research directions at the intersection of formal methods, differentiable physics, and AI-driven long-horizon robotics, aligning closely with NeurIPS’s mission to advance the frontiers of machine learning.

---

### Decision · Program_Chairs · 2025-09-17

**Decision:**

Reject

**Comment:**

This paper addresses the challenge of learning policies under linear temporal logic (LTL) specifications in reinforcement learning (RL), where sparse and discrete rewards make gradient-based optimization difficult. The authors propose a novel framework that introduces differentiable LTL rewards by (1) translating the LTL formula into an automaton (specifically LDBA) and modeling transitions probabilistically, (2) using soft labeling for atomic propositions, i.e., representing satisfaction as probabilistic values derived from distance measures, which are differentiable, and (3) constructing a product MDP between the environment and the probabilistic automaton, yielding a fully differentiable reward structure suitable for gradient-based optimization. The method is implemented using differentiable simulators and evaluated across standard benchmarks. Empirical results show that differentiable LTL rewards lead to faster convergence and higher-quality policies compared to traditional sparse-reward methods.

**Strengths**

All reviewers acknowledge that the core idea of constructing differentiable reward signals for LTL specifications via probabilistic automata and soft labeling is novel. This is particularly relevant for the intersection of formal methods and RL. Reviewers generally agree that the method is technically sound and well-aligned with the theoretical underpinnings of RL and formal logic. The paper presents extensive experiments demonstrating that differentiable rewards improve both convergence speed and final policy quality across low- and high-dimensional continuous control benchmarks. The ablation study effectively isolates the contribution of differentiability from environmental factors, further strengthening the experimental validation.

**Weaknesses**

The original submission provided formal guarantees only for the discrete LTL rewards (Theorem 1), with no similar justification for the differentiable case. This raised concerns about whether smooth reward signals preserve Büchi acceptance conditions in expectation or probabilistically. The benchmarks, while standard in differentiable RL, are relatively simple and short-horizon. Reviewers suggested evaluating on more challenging, long-horizon robotic tasks or stochastic domains to demonstrate broader applicability. Another recurring concern was the lack of sufficient baselines that compete in terms of structured reward propagation, such as reward machines.

During the rebuttal phase, the authors introduced a new theorem characterizing the discrepancy between discrete and differentiable LTL returns, thereby improving the formal rigor of the contribution. They also extended experiments to include reward machines as baselines. However, other recent RL for LTL baselines remain unaddressed. A large body of recent work exists on RL with LTL after reward machines, including potential-based reward shaping methods that exploit automaton structure, and techniques based on quantitative semantics for LTL/STL objectives. These methods also provide dense or shaped rewards aligned with logical specifications, making them direct competitors to the proposed approach. During the AC-reviewer discussion,  Reviewers expressed concern that since the rebuttal introduced a substantial set of changes, the paper would benefit from another round of review to ensure that these substantial additions are properly integrated, particularly the new theorem, which may warrant its own empirical validation.